# The Effect of Two Siderophore-Producing *Bacillus* Strains on the Growth Promotion of Perennial Ryegrass under Cadmium Stress

**DOI:** 10.3390/microorganisms12061083

**Published:** 2024-05-27

**Authors:** Lingling Wu, Yongli Xie, Junxi Li, Mingrong Han, Xue Yang, Feifei Chang

**Affiliations:** 1College of Agriculture and Animal Husbandry, Qinghai University, Xining 810016, China; ys210951310510@qhu.edu.cn (L.W.); m15565114991@163.com (J.L.); m18397002967@163.com (M.H.); m18097263030@163.com (X.Y.); 2017254@qhnu.edu.cn (F.C.); 2State Key Laboratory of Plateau Ecology and Agriculture, Qinghai University, Xining 810016, China; 3Key Laboratory of Use of Forage Germplasm Resources on Tibetan Plateau of Qinghai Province, Qinghai University, Xining 810016, China

**Keywords:** biological activity, cadmium contamination, Cd^2+^ tolerance, siderophore-producing *Bacillus*, ryegrass

## Abstract

Cadmium (Cd) is a highly toxic and cumulative environmental pollutant. Siderophores are heavy metal chelators with high affinity to heavy metals, such as Cd. Ryegrass (*Lolium perenne* L.) has a potential remediation capacity for soils contaminated by heavy metals. Consequently, using ryegrass alongside beneficial soil microorganisms that produce siderophores may be an effective means to remediate soils contaminated with Cd. In this study, the *Bacillus* strains WL1210 and CD303, which were previously isolated from the rhizospheres of *Nitraria tangutorum* in Wulan and *Peganum harmala* L. in Dachaidan, Qinghai, China, respectively, both arid and sandy environments, were evaluated for heavy metal pollution mitigation. Our quantitative analyses have discerned that the two bacterial strains possess commendable attributes of phosphorus (P) solubilization and potassium (K) dissolution, coupled with the capacity to produce phytohormones. To assess the heavy metal stress resilience of these strains, they were subjected to a cadmium concentration gradient, revealing their incremental growth despite cadmium presence, indicative of a pronounced tolerance threshold. The subsequent phylogenetic analysis, bolstered by robust genomic data from conserved housekeeping genes, including 16S rDNA, *gyr B* gene sequencing, as well as *dnaK* and *recA*, delineated a species-level phylogenetic tree, thereby confirming the strains as *Bacillus atrophaeus*. Additionally, we identified the types of iron-carrier-producing strains as catechol (WL1210) and carboxylic acid ferrophilin (CD303). A genomic analysis uncovered functional genes in strain CD303 associated with plant growth and iron carrier biosynthesis, such as *fnr* and *iscA*. Ryegrass seed germination assays, alongside morphological and physiological evaluations under diverse heavy metal stress, underscored the strains’ potential to enhance ryegrass growth under high cadmium stress when treated with bacterial suspensions. This insight probes the strains’ utility in leveraging alpine microbial resources and promoting ryegrass proliferation.

## 1. Introduction

Heavy metal pollution has become a global concern due to the acceleration of industrialization and continuous human population growth. Heavy metal elements, such as cadmium (Cd), lead (Pb), mercury (Hg), and copper (Cu), can have toxic effects on plant physiological functions when their concentrations exceed the plant’s tolerance threshold [1,2,3]. Although there is a growing awareness of the damage caused by heavy metals to plants, their important role in modern industry means that their production and use will continue to increase. Improving plant tolerance to heavy metals is therefore an important issue that needs to be addressed urgently [4].

Ryegrass (*Lolium perenne* L.) is an important grass plant with multiple functions and applications, especially in ecological restoration [5,6]. Li et al. [7] reported that *Bacillus subtilis* could adsorb Cd and improve Cd tolerance in *Poa annua* L. and *alfalfa* (*Medicago sativa* L.). Likewise, Liu et al. [8] demonstrated that the combined action of *Bacillus subtilis* and *Pseudomonas palustris* (Molisch) van Niel. effectively reduced Cd accumulation and promoted the growth and development of *Brassica chinensis* L. However, ryegrass has many limitations in the remediation of heavy metal pollution: 1. High concentrations of Cd^2+^ can significantly inhibit the growth of ryegrass. 2. Cd^2+^ stress causes a series of physiological and biochemical changes in ryegrass, such as an increase in the content of malondialdehyde (MDA), alteration of antioxidant enzyme activities (e.g., superoxide dismutase (SOD) and proline (PRO), etc. [9,10]. However, prolonged or high levels of Cd^2+^ stress may lead to the overloading of the antioxidant system, which may affect plant health and growth. Therefore, improving the tolerance of ryegrass under Cd^2+^ stress is more helpful for the efficiency of ecological restoration [11].

*Bacillus* spp. is a widely used inter-root growth promoter. Certain *Bacillus* species also exhibit high levels of tolerance to metal cadmium ions [6,12]. It has been reported that, under cadmium stress, *Bacillus* is able to promote the production of osmoregulatory substances, including proline, amino acids, and betaine, in plants, thereby maintaining osmotic pressure balance. At the same time, Cd stress causes the yellowing of plant leaves, which affects its normal photosynthesis, leading to a decrease in chlorophyll content and the inhibition of plant growth. Iron is one of the essential trace elements for plant growth and development, but it is often present in unavailable forms in the soil. Siderophores are a class of low-molecular-weight organic compounds that can efficiently chelate iron. By secreting siderophores, *Bacillus* can convert iron in the soil into a form that can be absorbed by plants, thus increasing the utilization of iron by plants. In addition, siderophores can also form stable chelates with heavy metal ions in the soil, thus reducing the bioavailability of these toxic metals and their toxic effects on plants [13]. For example, Wang et al. [14] observed that inoculating the *Bacillus* strains T1 and Y2 that produce siderophores into *Solanum nigrum* L. soils significantly promoted (*p* < 0.05) Cd absorption in soils and strengthened the ability of *Solanum nigrum* L. to remediate soils polluted by Cd.

Since *Bacillus* has the above-mentioned high-quality properties, we attempted to inoculate two strains of *Bacillus*, which were previously isolated, into Cd^2+^-stressed ryegrass and found that it had a growth-promoting effect on ryegrass under Cd^2+^ stress. Based on this, we hypothesized that the two strains of *Bacillus* might: 1. Promote the production of osmoregulatory substances by plants, thereby alleviating Cd^2+^ stress on ryegrass; 2. Have characteristic PGP activity, which exerts a growth-promoting effect on ryegrass; 3. Produce siderophores, which have a chelating effect on Cd^2+^, thereby alleviating stress on ryegrass.

## 2. Materials and Methods

### 2.1. Materials

The plant material used in this study was the ryegrass (*Lolium perenne* L.), variety “ESQUIRE 1” that originated in Denmark, purchased from Boya Flower Expo Center, Chengbei District, Xining City, Qinghai Province, China. The *Bacillus* strains WL1210 and CD303 were isolated from the rhizospheres of *Nitraria tangutorum* in Wulan, Qinghai, China, and *Peganum harmala* L. in Dachaidan, Qinghai, China, respectively. Both plants are present in arid and sandy environments. The primer synthesis was conducted at the Nanjing Jinshui Company, based in Beijing, China. Cadmium chloride (CdCl_2_) was procured from Zhan Yun Chemical Co., Ltd., located in Shanghai, China, while the ferrophilin test solution, also known as a CAS test solution, was sourced from Koolabo Technology Co., Ltd., based in Beijing, China. The Luria–Bertani (LB) medium was prepared according to the method described in [15]. The mannitol salt agar (MSA) medium was produced as previously described [16]. The silicate bacterial culture medium and the Monkina inorganic phosphorus bacterial medium were produced as described in [17].

### 2.2. Determination of Cd^2+^ Tolerance

The strains WL1210 and CD303 were cultured on a solid LB medium containing 20, 40, 60, and 80 mg/L Cd^2+^ for 1 d to observe their growth under Cd^2+^ stress [18,19]. CdCl_2_ was also added to the pre-prepared liquid LB medium at Cd^2+^ concentrations of 20, 40, 60, and 80 mg/L. The OD value at 600 nm was then determined based on turbidity using a liquid LB medium as the control (CK). Absorbances were measured every 6 h from 0 h onward until 48 h. The growth of the strains under Cd^2+^ stress was then observed.

### 2.3. Indole Acetic Acid Production and Gibberellin, and Cytokinin Capacity

To assess the indole acetic acid (IAA) production capacity, the strains WL1210 and CD303 were inoculated into 5 mL of liquid LB medium and incubated at a constant temperature, with shaking for 7 days. The strain cultures were then combined with the Salkowski coloritic solution (10 mL 0.5 mol/L FeCl_3_ + 30 mL distilled water + 50 mL 98% H_2_SO_4_) at a 2:1 volume ratio, mixed into a white ceramic plate, and allowed to stand in darkness for 30 min, followed by the evaluation of color change. The procedure was repeated in triplicate. A color change to red indicated that the strain produced IAA [20]. After 24 h of growth, the bacterial cultures were centrifuged (4 °C, 10,000 r/min) for 10 min. Then, 2 mL of supernatant was taken and placed in a centrifuge tube, followed by the addition of an equal amount of the colorimetric solution and the determination of OD_530_ values after being allowed to stand for 30 min in darkness. The amount of IAA produced by the strain was calculated according to the standard curve.

We referred to the gibberellin (GA) ELISA and cytokinin (CTK) ELISA detection kit for the reference. We cultivated the strains at 37 °C with 200 r/min shaking for 14 h. We transferred 1.5 mL to a 2 mL centrifuge tube and centrifuged it at 1000 r/min for 20 min. Then, we took 0.5 mL of the supernatant and processed it with the detection reagent from the kit, and then placed it in a 96-well plate. We measured the absorbance at OD450 using an enzyme-linked immunosorbent assay (ELISA) reader, constructed a standard curve, and calculated the concentration of GA and CTK in the strains [21].

### 2.4. Phosphate Solubilization and Potassium Release Capacity

The strain was activated in LB liquid medium and the activated strain was inoculated into the Montjuina inorganic phosphorus bacterial medium and silicate bacterial medium. Under sterile conditions, 5 μL of the strains were spotted onto a solid medium with 4 inoculation points per dish, one of which was a CK control (inoculated with LB liquid), and each strain was replicated 3 times. We cultivated them at a constant temperature of 28 °C for 7 days and 1–2 days, respectively. Based on the ratio of the phosphate-solubilizing zone diameter to the colony diameter (D/d), as well as the ratio of the potassium-solubilizing zone diameter to the colony diameter, its phosphorus and potassium solubilization abilities were determined. We observed and selected the colonies that showed phosphorus solubilization zones and oil droplet-like colonies for functional validation to determine their phosphorus solubilization and potassium release effects.

### 2.5. Determination of Siderophore-Producing Activity

#### 2.5.1. Quantitative and Qualitative Measurements

To identify the siderophore production by strains WL1210 and CD303, they were inoculated into 5 mL of liquid LB medium and grown for 24 h. The holes were cored on a solid Chrome Azurol S (CAS) medium, followed by the addition of the bacterial culture broth or sterile water (as the control) into the holes. After three days of cultivation at 30 °C, the presence of orange halos around the colonies was assessed.

Subsequently, the strains were inoculated into 5 mL of liquid mannitol salt agar (MSA) medium and grown at 30 °C, with shaking at 150 r/min for 48 h, followed by centrifugation at 6000 r/min for 10 min. Then, 1.5 mL of supernatant and 1.5 mL of CAS solution were added to a 5 mL centrifuge tube and thoroughly mixed. After being allowed to stand for 1 h, the absorption value (A) at 630 nm was determined, while deionized water was used as the control. In addition, 1.5 mL CAS solution was added to 1.5 mL of the liquid MSA medium without the bacterial culture and thoroughly mixed. The light absorption value was then determined as above and referred to as the reference ratio (Ar). The production of siderophores was calculated with the following standard formula (Equation (1)) and expressed as siderophore units (%SU). A/Ar was used as a quantitative index. Smaller ratios indicate greater yields of iron carriers. The general reference standard was A/Ar: 0–0.2, +++++; 0.2–0.4, ++++; 0.4–0.6, +++; 0.6–0.8, ++; and 0.8–1.0, + [18].
%SU (siderophore units) = (Ar − As)/Ar × 100(1)

In the formula, the terms are defined as follows:

%SU: This represents the percentage of siderophore units, which is a measure of the relative amount of siderophores produced by a microorganism;

(Ar − As): The equation calculates the difference in absorbance or biomass between two conditions;

Ar: This stands for the absorbance or biomass reading after the addition of an iron source or an inducer that promotes siderophore production;

As: The absorbance or biomass value in the absence of the iron source or inducer, serving as a control to compare to the Ar value.

#### 2.5.2. Type Identification of Chelating Groups in Iron Carriers

Ferritins can be classified into four types due to the different oxygen ligands bound to Fe^3+^, i.e., different chelating group structures: hydroxamic acid, catechol, hydroxycarboxylate, and mixed (i.e., including more than two chelating groups). The FeCl_3_ experiment, Arnow’s experiment, and Shenker’s experiment were used to determine the chelating group structure type of ferrophilin produced by the strain [15].
(1)FeCl_3_ experiment: We added 1−5 mL FeCl_3_ solution (2%) to 1 mL of supernatant. If it turned red or purple, there was ferrophilin, and if 1 mL of FeCl_3_ solution was added to 1 mL of supernatant and immediately turned red, it was of the hydroxamic acid type. If 1 mL of supernatant became red or purple after adding more than 1 mL of FeCl_3_ solution, it was catecholic.(2)Arnow’s experiment: A total of 1 mL of 0.5 mol/L HCl and 1 mL of 10% sodium molybdate-sodium nitrite solution were added to 1 mL of supernatant, and if there was a catechol structure in the solution, the nitrite decomposed to form a yellow ligand, and the solution turned yellow. We continued to add 1 mL of NaOH (1 mol/L) solution, which turned red (does not change color for at least 1 h) if it contained catechol ferrophile, and has a characteristic absorption peak at 510 nm on a UV spectrophotometer.(3)Shenker’s experiment: A total of 1 mL of CuSO_4_ (750 μmol/L) solution and 2 mL of acetate buffer (pH = 4.0) were added to 1 mL of supernatant. The wavelengths in the range of 190−280 nm were scanned in a UV spectrophotometer to see if there were corresponding absorption peaks.

### 2.6. Phylogenetic Analysis

16S rDNA amplification. PCRs were conducted with the forward primer 27F: 5′-AGAGTTTGATCMTGGCTCAG-3′ and reverse primer 1492R: 5′-GGYTACCTTGTTACGACTT-3′. *gyr B* gene amplification. We used the forward primer UP2r: 5′-AGCAGG-GTACGGATGTGCGAGCCRTCNACRTCNGCRTCNGT-CAT-3′ and reverse primer UP1: 5′-GAAGTCATCATGAC-CGTTCTGCAYGCNGGNGGNAARTTYGA-3′.

We utilized the 16S rDNA and *gyr B* genes to perform sequence alignments for the strains WL1210 and CD303 on NCBI, while also incorporating whole-genome data to compare the housekeeping sequences of *dnaK*, *recA*, and *rpoB* for strain CD303.

The nucleotide sequences were analyzed with the BLASTn https://blast.ncbi.nlm.nih.gov/Blast.cgi (accessed on 3 April 2024) search algorithm in the GenBank database of the National Center for Biotechnology Information (NCBI) and then aligned to their nearest neighbors using highly similar sequences (megablast). The phylogenetic trees were constructed with the maximum likelihood (ML) and neighbor-joining (NJ) methods with 1000 bootstraps using the MEGA 7.0 software. The edited sequences were submitted to GenBank (NCBI).

### 2.7. Effect of the Strains on Ryegrass Growth under Cd^2+^ Stress

#### 2.7.1. Determination of Cd^2+^ Effects on Ryegrass Seed Germination

Ryegrass seeds of uniform size and that were full were selected and disinfected with a 20% sodium hypochlorite solution for 30 min. They were then repeatedly rinsed with deionized water 3–4 times, and CdCl_2_ solutions of 0 (CK), 20, 40, 60, and 80 mg/L were prepared. After high-temperature sterilization, 5 mL of the CdCl_2_ solutions of the same volume but different concentrations were added to a clean Petri dish to saturate the filter paper. The seeds were seeded in the Petri dish, with 30 seeds treated per group, and triplicate dishes per group. A sterile water treatment was used as the control and was cultured at room temperature (21 °C), with a light/dark period of 12/12 h [22].

The number of germinated seeds in each treatment was recorded every day until the seeds no longer germinated. The germinated seedlings were gently removed with tweezers, followed by the determination of seedling heights and root lengths. Survey data were used to determine germination potential, germination rate, germination index, and the vitality index of treated seeds, as previously described [23,24]. The germination potential was calculated by (GV,%) = (number of germinated seeds after 7 days/number of tested seeds) × 100%. The germination rate was calculated as (GR,%) = (number of germinated seeds after 10 days/number of tested seeds) × 100%. The germination index was calculated as (GI) = ΣGt/Dt, where Gt is the number of germinated seeds at time t and Dt is the corresponding days of germination. Finally, the vitality index was calculated as (VI) = GI × S, where GI is the germination index and S is the seedling length for a certain period.

#### 2.7.2. The Effect of the Strain on Ryegrass Growth under Cd^2+^ Stress

To evaluate the effect of the strain inoculation on ryegrass growth due to Cd^2+^ stress, a *Bacillus* suspension was prepared and cultured in 5 mL of liquid LB medium for 12–14 h at 37 °C and with shaking at 180 r/min. The culture was then transferred to a 50 mL centrifuge tube and centrifuged at 10 °C and 8000 r/min for 10 min. The cell pellet was aseptically washed three times, and 50 mL of sterile water was added to generate a bacterial suspension. LB liquid alone was used as the control. The OD_600_ values were measured and cell concentrations were adjusted to 1 × 10^7^ CFU/mL (OD_600_ = 1.0) [25].

To evaluate the effects of the strains on ryegrass growth, natural soil and vermiculite were mixed at a 2:1 ratio and sterilized by autoclaving at 121 °C for 1 h, followed by laying soils on plastic. The prepared Cd^2+^ solutions at different concentrations were sprayed on the sterilized soil, evenly mixed, and allowed to sit for 15 days to achieve heavy metal stabilization. To allow Cd^2+^ to fully absorb into soils to prevent leaching loss in later watering, sterilized soil was used as the control (CK) and ryegrass seeds were soaked in a 20% sodium hypochlorite solution for 30 min and rinsed with sterile water.

Thirty disinfected ryegrass seeds were randomly selected and planted in 0.35 L flowerpots with a diameter of 10 cm and a height of 10 cm for pot experiments, followed by culturing in an intelligent artificial climate box (25 °C, light/dark period of 16/8 h). When the seedling heights reached 5–10 cm, the plants were irrigated with 50 mL of a bacterial suspension once every 3 days, while the control group was treated with the same volume of sterile water. After 20 days of cultivation, the seedlings were removed for a root washing treatment, and 15 ryegrass seedlings from each group under different treatments were used to measure indicators such as plant height, root length, and fresh weight [26].

To evaluate the effects of the two strains on the promotion of ryegrass under different concentrations of cadmium stress, we established two treatments: the first treatment group: CK (sterile water), CK + W/CK + C (bacterial suspension), 40 (40 mg/L Cd^2+^ stress), 60 (60 mg/L Cd^2+^ stress), and 80 (40 mg/L Cd^2+^ stress); and the second as the control: 20 + W/20 + C (20 mg/L Cd^2+^ stress + WL1210/CD303 irrigation), 40 + W/40 + C (40 mg/L Cd^2+^ stress WL1210/CD303 irrigation), 60 + W/60 + C (60 mg/L Cd^2+^ stress + WL1210/CD303 irrigation), and 80 + W/80 + C (80 mg/L Cd^2+^ stress + WL1210/CD303 irrigation).

#### 2.7.3. The Effects of the Strain on the Physiological Indicators of Ryegrass under Cd^2+^ Stress

##### Measurement of the Chlorophyll Content

The experimental guide of plant physiology was used as a reference for measuring chlorophyll [26]. A total of 0.4 g of ryegrass leaves were placed in a mortar, and a small amount of quartz sand was added, in addition to calcium carbonate powder and 5 mL of 95% ethanol, followed by grinding the mixture to a homogenate. A small amount of ethanol was added, and the mixture was ground until the tissue turned white, and then it was allowed to sit for 30 min. The mixture was then strained into a 50 mL brown volumetric bottle, the mortar and glass rod were washed with ethanol, and the bottle was filled with 95% ethanol up to 50 mL. This process was repeated twice. The solution was poured into a colorimetric dish, and the absorbances were measured at wavelengths of 663, 645, and 470 nm. Each replicate was measured three times at different wavelengths, and the values were averaged. The concentrations of chlorophyll a and chlorophyll b were added together to obtain the total chlorophyll concentration using the formulas below (Equation (2)).

The calculations of the pigment concentration in 95% ethanol extract are as follows:C_a_ (chlorophyll a) = 13.95A_665_ − 6.8A_649_
 C_b_ (chlorophyll b) = 24.96A_649_ − 7.32A_665_(2)
    C_T_ (chlorophyll)= C_a_ + C_b_ = 18.16A_649_ + 6.63A_665_

##### Measurement of the Proline Content

A plant proline ELISA detection kit was used to measure proline concentrations. Briefly, 0.05 g of fresh ryegrass leaves were added to 0.5 mL of the extract solution and ground to a homogenate in an ice bath. The mixture was then extracted in a boiling water bath for 10 min and centrifuged at room temperature for 10 min at 10,000 r/min. The supernatant was then measured and allowed to cool before measurement. A standard curve of the equation was used to calculate the measured proline concentrations. The measurements were calculated according to sample fresh weight as Pro(μg/g fresh weight) = Y × V inverse total ÷ (V sample ÷ V sample total × W) = 3XY ÷ W, where W refers to the sample quality.

##### Measurement of the Protein Content

Protein concentrations were detected using a BCA protein content kit. Briefly, 0.05 g of ryegrass leaves were added to 0.5 mL of distilled water in an ice bath, followed by grinding to a homogenate and centrifuging at 12,000 r/min at 4 °C for 10 min. The supernatant was then used for measurement. Standard curves were prepared and used for calculations. Sample fresh weight calculations were conducted using the equation: Cpr fresh weight (mg/g) = [(Y + 0.0046) present 0.188 × 10^−3^] present (V1 present V (W) × D = 1.06 ×(Y + 0.0046) present W × D, where W is the sample quality and D is the dilution ratio, with undiluted defined as 1.

##### Measurement of the Superoxide Dismutase Content

The SOD enzyme activity was determined using a kit. Weigh 0.1 g of the fresh plant sample and add 1 mL of pre-cooled 1 × lysis buffer for ultrasonic crushing. After ultrasonic crushing for 3 min (30% power, 3 s, 7 s interval), centrifuge at 12,000× *g* for 5 min at 4 °C, and then take the supernatant for testing. After mixing well and incubating at 37 °C for 30 min, measure the absorbance values of each well at 450 nm and calculate them.

### 2.8. Determination of Cadmium in Soil and Plants

Cadmium Determination in Soil and Plant Samples by ICP-MS Following Multi-Acid Digestion.

Soil Analysis: Accurately weigh 0.1 g of the soil sample to transfer it into a digestion vessel. Add 5 mL of concentrated nitric acid to the vessel and allow the sample to stand overnight. Subsequently, add 2 mL of hydrogen peroxide and 2 mL of hydrofluoric acid to the vessel, secure it with a stainless steel jacket, and then place it in a constant temperature oven for digestion at the temperature of 150–170 °C for a period of 9 h. After the digestion is complete and the vessel has cooled, carefully loosen and remove the stainless steel jacket, and take out the digestion vessel. Evaporate the solution to near dryness on a hot plate at a temperature of 160 °C for approximately 30 min. Make up the volume to 25 mL with a 1% nitric acid solution and proceed to ICP-MS analysis for the determination of the elemental content.

Plant Analysis: Accurately weigh 0.2–1.0 g of dry or 1.0–2.0 g of fresh plant material to transfer it into a polytetrafluoroethylene (PTFE) digestion vessel. Add 5 mL of nitric acid to the vessel and allow the sample to soak overnight. Close the vessel with a lid, secure it with a stainless steel jacket, and then place it in a constant temperature oven. Heat the sample at 80 °C for 1–2 h, increase the temperature to 120 °C for another 1–2 h, and finally raise it to 160 °C for 4 h. Allow the vessel to cool naturally to room temperature within the oven. After cooling, evaporate the acid to near dryness on a hot plate. Transfer the digested solution to a 25 mL volumetric flask, and rinse the digestion vessel and its lid three times with 1% nitric acid, combining all rinses into the flask. Make up the volume to the mark with 1% nitric acid, mix well, and set the solution aside for analysis.

A reagent blank should also be processed concurrently for quality control purposes. The prepared solution is then analyzed using ICP-MS to measure the content of cadmium.

### 2.9. Whole-Genome Sequencing

Genome sequencing was conducted for the strain *Bacillus* CD303 inoculated into 5 mL of the liquid LB medium that was incubated at 37 °C, with shaking at 200 r/min for 14 h, followed by centrifugation at 10,000 r/min for 10 min. The resultant cell pellet was quick-frozen in liquid nitrogen. The genomic DNA was extracted using the Wizard^®^ Genomic DNA Purification Kit (Promega, Tokyo, Japan), according to the manufacturer’s protocol. The purified genomic DNA was quantified by TBS-380 fluorometer (Turner BioSystems Inc., Sunnyvale, CA, USA). High-quality DNA (OD_260/280_ = 1.8~2.0, >20 μg) was used to conduct further research. The genome was sequenced using a combination of the PacBio RS II Single MoleculeReal Time (SMRT) and Illumina (San Diego, CA, USA) sequencing platforms. For Illumina sequencing, at least 1 μg genomic DNA was used for each strain in the sequencing library construction. The DNA samples were sheared into 400–500 bp fragments using a Covaris M220 Focused Acoustic Shearer (Covaris, Inc., Woburn, MA, USA). following the manufacturer’s protocol. The Illumina sequencing libraries were prepared from the sheared fragments using the NEXTflex™ Rapid DNA-Seq Kit (Scientific Services, Inc., Cary, NC, USA). Briefly, 5′ prime ends were first end-repaired and phosphorylated. Next, the 3′ ends were A-tailed and ligated to sequencing adapters. The third step was to enrich the adapter-ligated products using a PCR. The prepared libraries then were used for paired-end Illumina sequencing (2 × 150 bp) on an Illumina HiSeq X Ten machine (San Diego, CA, USA). For Pacific Biosciences sequencing, an aliquot of 15 μg DNA was spun in a Covaris g-TUBE (Covaris, Woburn, MA, USA) at 6000 r/min for 60 s using an Eppendorf 5424 centrifuge (Eppendorf, Hauppauge, NY, USA). The DNA fragments were then purified, end-repaired, and ligated with SMRTbell sequencing adapters, following the manufacturer’s recommendations (Pacific Biosciences, Menlo Park, CA, USA). The resulting sequencing library were purified three times using 0.45 × volumes of Agencourt AMPure XP beads (Beckman Coulter Genomics, Danvers, MA, USA), following the manufacturer’s recommendations. Next, a ~10 kb insert library was prepared and sequenced on one SMRT cell using the standard methods. Glimmer (University of Washington) was used for CDS prediction, tRNA-scan-SE was used for tRNA prediction, and Barrnap was used for rRNA prediction. The predicted CDSs were annotated from the NR, Swiss-Prot, Pfam, GO, COG, and KEGG databased using sequence alignment tools, such as BLAST, Diamond, and HMMER. Briefly, each set of query proteins were aligned with the databases, and annotations of the best-matched subjects (e-value < 10^−5^) were obtained for gene annotation.

### 2.10. Statistical Analysis

The plant indexes of the different treatments were analyzed by a one-way analysis of variance (ANOVA) in SPSS 25. The level of significance among the different treatments was tested at *p* < 0.05 using multiple comparisons (LSD test). Before the data analysis, the single sample Kolmogorov–Smirnov test was used to test the normality of the data to indicate whether the data followed a normal distribution and the graphs were drawn by Origin (2021). The tRNAscan-SE v2.0 software http://trna.ucsc.edu/software/, (accessed on 9 May 2024) was used to predict the tRNA contained in the genome. The phylogenetic tree was constructed using the software MEGA 7.0.

## 3. Results

### 3.1. Cd^2+^ Tolerance of the Strains

The Cd^2+^ tolerance levels of the strains WL1210 and CD303 were determined. The strains WL1210 and CD303 grew slowly after 24 h of culture on the solid LB medium with Cd^2+^ concentrations of 20, 40, 60, and 80 mg/L. The absorbance values at 600 nm were measured every 6 h under Cd^2+^ exposure. Increases in Cd^2+^ concentration caused a gradual slowing of the strain growth, indicating that Cd^2+^ stress inhibited growth. Nevertheless, the two strains grew under different Cd^2+^ concentrations and exhibited a slow growth over 48 h. Thus, the two strains exhibited some tolerance to different Cd^2+^ concentrations (Figure 1).

### 3.2. Strains’ Ability to Produce CTK, GA, and IAA

The standard curves of CTK and GA were made, and the standard equations were obtained: Y = 0.0419X + 0.0683 and Y = 4.762X + 0.0673. It was found that the yields of CTK and GA of the strains WL1210 and CD303 were 5.2, 4.3, 3.5, and 4.8 nmol/L, respectively.

The IAA production abilities of the strains were qualitatively and quantitatively measured. The strains caused the Salkowski solution to turn red, indicating IAA production capacity. The equation for the standard curve was y = 0.032x + 0.019, while the absorbance values of the strains WL1210 and CD303 at 530 nm were 0.276 and 0.241, respectively. Thus, the IAA concentrations secreted by the strains WL1210 and CD303 were 15.03 and 17.94 mg/L, respectively, consequently indicating a good IAA production capacity (Figure 2b).

### 3.3. Phosphorus Solubilization and Potassium Release Activity Assay

Phosphorus and potassium are essential nutrients for plant growth and play an extremely important role in the growth and development of plants. The bacterial suspension of the strains was inoculated on the organophosphorus medium of Monascus for 3 d, and it was found that both strains could dissolve organophosphorus to different degrees. The strains WL1210 and CD303 had phosphorus-solubilizing halo (D) to colony diameter (d) ratios, A (D/d), of 3.27 and 3.46, respectively. The strains WL1210 and CD303 had potassium-solubilizing halo (D) to colony diameter (d) ratios, A (D/d), of 1.27 and 1.46, respectively. The strains were inoculated on a plate of silicate bacterial medium and cultured in a constant temperature incubator for 1–2 d, and it was observed that there were oil droplet colonies in the plates of the two strains, which indicated that the two strains had a better function of potassium solubilization (Figure 3).

### 3.4. Siderophore-Producing Activities

Color changes due to the strains WL1210 and CD303 on CAS media were observed, with light yellow halos identified around the colonies (Figure 2a). The OD values for the strains WL1210 and CD303 at 630 nm were 1.545 and 1.213, respectively. Consequently, the ferrophilin activity values of the strains WL1210 and CD303 were 36.2 and 49.9%, respectively (Table 1), indicating that both strains exhibited a good siderophore production ability.

### 3.5. Identification of the Type of Ferrophilin-Chelating Groups

Using the Arnow method [27], the WL1210 solution turned red immediately after NaOH was added to the fermentation supernatant, and an absorption peak was detected at 510 nm. The ferrophilin produced by the strain WL1210 was identified as catechols (Figure 2c). The Shenker method [27] was used to detect the maximum absorption peak of the strain CD303 at 190–280 nm; so, the degree of the strain CD303 was identified as carboxylic acid ferrophilin (Figure 4).

### 3.6. Phylogenetic Analysis

Phylogenies based on the sequences of 16S rDNA and *gyr B* were powerful enough in discriminating most *Bacillus* isolates to the species level, but came of short in resolving the exact identity of a few isolates, making the utilization of additional housekeeping genes like *rpoB*, *recA*, and *dnaK* in those exceptional cases indispensable. The homology results indicate that the strains WL1210 and CD303 are both classified as *Bacillus atrophaeus* (Figure 5).

### 3.7. Effect of the Strains on Ryegrass Growth Promotion under Cd^2+^ Stress

#### 3.7.1. Effects of Cd^2+^ Stress on Ryegrass Seed Germination

The germination rate of ryegrass seeds under Cd^2+^ stress was determined. Compared to the control group (CK), the seed germination rate was less affected by Cd^2+^ following exposure to concentrations of 20, 40, and 60 mg/L for 7 days of cultivation, with the germination rates remaining above 75%. After the 10th day, the germination rate remained above 80%, while the GI and VI were above 4.8 and 8.7, respectively. At 80 mg/L Cd^2+^ concentration, the germination rates of the seeds were significantly inhibited. The seed germination rate of ryegrass after 7 days was only 18%. After 10 d, the germination rate remained above 34.67%, while the GI and VI were only about 1.4. Thus, the ryegrass seeds exhibited Cd tolerance and could normally germinate when the Cd^2+^ concentrations were low. However, at Cd^2+^ concentrations of 80 mg/L, toxicity resistance was reduced and seed germination was limited (Table 2 and Table 3). Values are means + SE (n = 10). The variants in the same column marked with different letters represent the mean values that are statistically different from each other according to the Duncan’s test (n = 10, *p* < 0.05)

#### 3.7.2. Effects of the Strains on Ryegrass Biomass under Cd^2+^ Stress

The germination experiments revealed that ryegrass seed germination was inhibited due to 80 mg/L Cd^2+^ stress. Under different concentrations of Cd^2+^ stress, WL1210 and CD303 strain suspension inoculation significantly promoted ryegrass seedling growth. Compared to the control group (CK), after root irrigation with the strains WL1210 and CD303, the plant heights and root lengths of ryegrass increased by 27.7 and 13.6, and 44.7 and 35.1%, respectively. They all showed significant differences. The plant height and root length of ryegrass were better than those of ryegrass treated with cadmium stress only at different concentrations, indicating that the addition of a bacterial suspension could promote the growth of ryegrass to a certain extent. The 20 mg/L Cd^2+^ concentration showed a significant difference in plant height, indicating that cadmium stress at low concentrations had a promotional effect on the growth of ryegrass, and a certain cadmium ion stress can promote root growth. After adding the WL1210 and CD303 strain suspensions (80 + W and 80 + C) under 80 mg/L Cd^2+^ stress, ryegrass growth maintained some Cd tolerance and was less affected by Cd^2+^ stress relative to the controls. A strong chelation effect has been observed between siderophores and heavy metal ions in recent years, with siderophore-producing strains generally having a higher tolerance to heavy metals, further supporting that the two strains promote the growth of ryegrass under Cd stress (Figure 6).

#### 3.7.3. Effects of the Strains on the Chlorophyll Contents of Ryegrass under Cd^2+^ Stress

Compared to the control (CK), the chlorophyll contents of ryegrass increased by 80% and 77.8% after treatment with the WL1210 and CD303 suspensions, respectively. The protein contents of ryegrass with the same volume of bacterial suspension added at 20 and 40 mg/L Cd^2+^ concentrations were elevated and significantly different from the chlorophyll content under cadmium stress treatment only when compared to the control, whereas the chlorophyll content was elevated at the 60 and 80 mg/L concentrations when compared to the control, but there was no significant difference. Cd stress inhibited plant photosynthesis. After adding the WL1210 and CD303 strain suspensions under 80 mg/L Cd^2+^ stress, the chlorophyll contents decreased compared to those of the control (CK), although the effect was small. Furthermore, the plants maintained a relatively normal photosynthesis activity, indicating that the two strains significantly improved plant resistance to toxicity (Figure 7a,b).

#### 3.7.4. Effects of the Strains on the Proline Content of Ryegrass under Cd^2+^ Stress

Proline concentrations in plants reflect their stress resistance. A standard proline curve was established by the equation: Y = 164.75X + 8.3415. Compared to control (CK), plant proline content increased after the addition of the bacterial suspension inoculations (WL1210 and CD303) by 23.6 and 35.9%, respectively. Under different concentrations of Cd^2+^ stress, and after adding the same volume of bacterial suspensions, the proline content also increased in all. At 80 mg/L Cd^2+^ stress, the proline contents of the WL1210 and CD303 bacterial suspension-treated plants after root administration were up to 46.85 and 56.88 μg/g, respectively. Thus, the ryegrass proline contents under Cd stress increased and inoculation with the strains WL1210 and CD303 can reduce Cd stress damage to ryegrass through cellular osmosis regulation and enhances plant stress resistance (Figure 7c,d).

#### 3.7.5. Effect of Cd^2+^ and Strain Inoculation on Ryegrass Protein Concentrations

Proteins play a crucial role in plant enzymatic processes, ensuring the stability and efficient functioning of cellular metabolism. A standard equation was obtained for protein quantification: Y = 0.188X − 0.0046. Compared to the control (CK), the protein contents of ryegrass were significantly higher by 80% and 77.8%, respectively, after inoculation with the bacterial suspensions (WL1210 and CD303, respectively), representing significant differences from the control (CK). Under different concentrations of Cd^2+^ stress, the protein concentrations increased when considering the pairwise comparisons, indicating that the two strains promoted the growth of ryegrass seedlings and maintained normal plant growth under Cd stress. These results reflect the indirect growth promotion effects of the strains (Figure 7e,f) based on a *t*-test.

#### 3.7.6. Effect of Cd^2+^ and Strain Inoculation on Ryegrass Superoxide Dismutase (SOD) Concentrations

Under heavy metal stress, plants increase the synthesis of SOD by regulating the expression of SOD genes, thereby enhancing its activity. This up-regulation is a physiological response to plant adaptation to heavy metal stress and helps to enhance plant tolerance to oxidative stress. The increase in SOD activity helps to scavenge excess superoxide anion and reduce subsequent oxidative damage. The SOD content was elevated after root irrigation with the fungal suspension compared to the control, and its SOD content was significantly higher after root irrigation with the fungal suspension at 20, 40, and 60 mg/LCd^2+^, suggesting that ryegrass is better able to maintain intracellular redox homeostasis by increasing the activity of SOD under Cd^2+^ stress, thus protecting the cells from oxidative damage and enhancing its resistance to heavy metal stress (Figure 7g,h).

### 3.8. Determination of Cadmium in Soil and Plants

To determine the effects of the inoculation strains WL1210 and CD303 on Cd^2+^ accumulation in ryegrass seedlings and the enrichment of soil cadmium, the total cadmium concentrations in soil, ryegrass roots, and leaves were measured. Compared to the control, after irrigation with 20 mg/L Cd^2+^ and the bacterial suspension, the total cadmium content in the soil, ryegrass roots, and leaves was the lowest, showing extremely significant differences. After irrigation with 40 mg/L Cd^2+^ and the bacterial suspension, the total cadmium content in the soil, ryegrass roots, and leaves showed significant differences compared to the control. After irrigation with 60 mg/L Cd^2+^ and the bacterial suspension, the total cadmium content in the soil, ryegrass roots, and leaves also showed differences. Except for the soil treated with the bacterial suspension WL1210, especially under 60 mg/L Cd^2+^, the toxic Cd^2+^ accumulated the most in the soil, ryegrass root system, and leaves (Figure 8).

### 3.9. Genomic Characteristics of Strain CD303

The whole genome of the strain CD303 comprised a circular chromosome of 4,288,753 bp in length (Figure 9). The average GC content was 43.19%, the number of total reads was 54,544, and the total base length was 608,867,443 bp. A total of 4084 coding sequences were identified, and the total length of coding genes was 3,716,880 bp, while the average coding sequence length was 910.11 bp. The genome encoded 24 rRNA genes, including 8 16S, 8 23S, and 8 5S rRNA genes, in addition to 82 tRNAs and 89 sRNAs. A total of 38 tandem repeats were identified, covering a total length of 13,852 bp. It was logged as CP135964.

#### 3.9.1. Growth-Promoting Functional Genes

The gene functional annotation analysis revealed the presence of *glpQ*, *hxlA*, and *hxlB* in the CD303 genome, which are key genes involved in growth hormone synthesis and the tryptophan-dependent IAA synthesis pathway of *Bacillus*. Numerous studies have shown that *Bacillus* can synthesize IAA, which promotes plant growth. In addition, the genes *miaA* and *miaB* related to CTK synthesis were identified in the genome. CTK can participate in plant growth and development, in addition to improving plant tolerance to abiotic stress. In addition, the CD303 genome encodes various functional genes involved in the direct growth promotion of *Bacillus*, including *ndhF*, *frmA*, *phnB*, *glnA*, and *glnR*, which are involved in the synthesis of nutrients required for plant growth and development, nitrogen fixation, phosphorus dissolution, and other processes that improve soil environments, thereby promoting plant growth. *fnr* and *iscA* encode proteins related to the synthesis and transport of iron carriers that can increase the iron content of plants by chelating iron. Consequently, these iron-producing carrier proteins can indirectly promote the growth of plants by binding heavy metal ions, thereby reducing the toxicity of metals in plants (Table 4).

#### 3.9.2. Functional Annotation of the CD303 Gene

COG annotation results: The genome sequence of the strain CD303 was compared to the egg NOG database for COG functional annotation. A total of 3258 genes were annotated, accounting for 79.77% of all genes, and four classifications and 23 types were performed. Among the predicted known functional genes, the number of genes related to the functions of amino acid transport and metabolism, carbohydrate transport and metabolism, general function prediction and translation, ribosome structure and biogenesis, and signal transduction mechanism was high (Figure 10).

#### 3.9.3. Stress Tolerance-Related Functional Genes

The CD303 genome also encodes key genes involved in inducing plant stress resistance. For example, *glmS* and *tarS* can improve mechanical strength, help to maintain cell shape, and assist cell movement and growth. A gene cluster was identified that encodes lipopeptide antibacterial proteins, including *LicA*, *LicB*, *LicC,* Pangraptin, and surfactant proteins that can significantly inhibit the growth of *Fusarium fungi.* Furthermore, the CD303 genome harbored functional genes encoding osmoregulatory proteins involved in responses to abiotic stress. Specifically, the gene cluster containing *mgtC* was encoded, which regulates intracellular pH, maintains ionic homeostasis, and promotes cell growth under stress. Furthermore, *proB* and *proC* were identified, which encode proline and betaine osmotic proteins that can increase tolerance to osmotic stress (Table 4).

## 4. Discussion

The soil, a critical non-renewable resource, is under threat from heavy metal contamination caused by human activities [28,29], with phytoremediation and microbial remediation strategies emerging as preferred solutions in the restoration industry due to their sustainability and cost-effectiveness [29,30,31,32,33].

Plant-growth-promoting rhizobacteria (PGPRs), specifically *Bacillus* spp., possess intrinsic characteristics that are instrumental in promoting plant growth and safeguarding against environmental stressors. As a natural source of phytohormones, PGPRs may produce a variety of phytohormones concurrently, which is crucial for promoting root development, nutrient uptake, biomass synthesis, and other characteristics of the host plants [34]. Phytohormones secreted by PGPRs, like auxin (IAA), cytokinin (CTK), and gibberellin (GA) [35], make a significant contribution to plant growth under heavy metal stress conditions [36].

In this study, we assessed the bioactivity and stress tolerance of the superior strains WL1210 and CD303 and discovered that, under Cd stress, both strains exerted a growth-promoting effect on ryegrass. Our findings indicate that the strains WL1210 and CD303 exhibit excellent tolerance to Cd^2+^ and are capable of producing plant hormones, such as indole-3-acetic acid (IAA), cytokinins (CTKs), and gibberellins (GAs). Many PGPRs, including *Bacillus* spp., *Pseudomonas* spp., *Xanthomonas* spp., *Rhizobium* spp., *Arthrobacter* spp., and *Bradyrhizobium* spp., have been reported to synthesize IAA. In heavy metal pollution areas, PGPRs can stimulate plant root growth by secreting IAA, which increases the plant root surface area and enhances access to soil nutrients and heavy metal accumulation, so as to improve the efficiency of the plant root absorption of heavy metals. Gavrilescu et al. reported that stress from Pb, Cu, and as significantly increased the amount of IAA synthesized by *Serratia* K120, *Enterobacter* K131, *Enterobacter* N9, and *Escherichia coli* N16, promoting the growth of sunflower plants, which has a potential application in phytoremediation systems [37]. Cytokinin (CTK) is a hormone widely found in higher plants, algae, and bacteria [38], and is the second most important phytohormone after IAA [39]. Currently, numerous studies have shown that PGPRs can produce cytokinins and promote plant growth under heavy metal stress [39]. For example, Li et al. inoculated *Arabidopsis thaliana* with a cytokinin-producing PGPR, *Bacillus megalosporum*, and found that elevated transcript levels of the cytokinin receptors in plant root shoots and roots significantly promoted plant growth [40]. Bioactive gibberellin (GA) is a diterpenoid plant hormone that is involved in several plant developmental processes, including seed dormancy, germination, flowering, fruit ripening, root growth, and root hair enrichment, through complex biosynthesis [41,42]. At present, more than 130 kinds of GAs have been found. PGPRs mainly produce GA1, GA3, GA4, and GA20, with GA3 being the most prevalent form [43]. Typical PGPR species that can produce GA include *Acetobacter* spp., *Bacillus* spp., and *Azotobacter* spp. [44]. GA can improve plant adaptation to heavy metal toxicity. One study showed that the application of GA3 alleviated the toxicity of Cu stress in pea (*Pisum sativum* L.) seedlings [45]. Under Cd stress, GA application on *Cyphomandra betacea* promoted beetroot seedling growth and increased the biomass, leaf net photosynthetic rate, and carotenoid and soluble sugar contents. The Cd content of *C. betacea* seedlings reduced gradually with the increase in the concentrations of GA. For the hyperaccumulator *Sedum alfredii,* the application of GA significantly increased the dry biomass of the roots, stem, leaves, and shoots. The enhanced accumulation of Cd and Pb in the shoots of *S. alfredii* demonstrates a significant potential for heavy metal phytoremediation [46].

Nitrogen is not only one of the most important nutrition sources for plant growth, flowering, and fruiting, but also an essential element of amino acids, protein, chlorophyll, nucleic acids, membrane lipids, ATP, NADH, co-enzymes, etc. [47,48]. PGPRs contain the key enzyme nitrogenase of BNF, with its activity typically regulated by the transcription of the nitrogen fixation gene (*nif*) [49]. Nitrogen fixation is instrumental in promoting plant growth, highlighting one of the vital roles of PGPRs in enhancing plant vitality. Six common PGPR strains exhibited nitrogenase activity and significantly promoted the growth of wheat and spinach in the study by Santos et al. [50]. It was also verified that the inoculation of PGPRs significantly increased the nitrogenase activity of *Dalbergia sissoo* seedlings and promoted its growth [51]. Phosphorus stands as a crucial nutrient vital for plant metabolism and nutrient cycling, ranking second only to nitrogen in significance [52,53]. One of the most promising avenues for developing durable and secure technology is the application of phosphorus-solubilizing bacteria (PSBs) [54], which play an important role in phosphorus cycling and promoting plant growth. It has been demonstrated that PGPRs function as PSBs to transform insoluble phosphorus into a form that plants can use [55], mainly by secreting organic acids (like citric acid, oxalic acid, and gluconic acid), chelating metal ions to form soluble complexes (such as phosphates of calcium, iron, and aluminum), and by producing enzymes (pyridoxal phosphatase (PDXP), phytase, and C-P lyase) to hydrolyze organic phosphorus in soil into inorganic forms. In the course of this investigation, it was revealed that the duo of Bacillus spp. strains demonstrated a capacity to solubilize phosphorus and potassium, a function that significantly benefits the plants’ ability to assimilate these essential nutrients. These results serve to highlight the strains’ considerable potential for advancing the uptake of nutrients by plants and for the enhancement in soil fertility. The strains’ solubilization capabilities augment the bioavailability of phosphorus and potassium within the soil matrix, facilitating their uptake by plants and consequently stimulating vegetative growth and crop yield. Furthermore, this intrinsic biological function may contribute positively to the broader objectives of sustainable agriculture, particularly by diminishing the reliance on chemical fertilizers and by bolstering the crops’ inherent resilience to environmental stressors. Meanwhile, by irrigating ryegrass with bacterial suspensions containing these strains under different treatments, we observed that the plants were less affected by Cd^2+^ stress, promoting their growth. Concurrently, the contents of chlorophyll, proline, and protein in ryegrass increased, indicating that both strains exert a growth-promoting effect on ryegrass under cadmium stress (Figure 11). After ryegrass was inoculated with the bacterial suspension, the cadmium content in soil, plant roots, and leaves was measured. It was found that the two strains of siderophore-producing *Bacillus* had an obvious degradation effect on the heavy metal cadmium under 20, 40, and 60 mg/L Cd^2+^ stress, and the degradation effect was more significant under 20 and 40 mg/L Cd^2+^ stress. This may be related to the growth of the strain at this concentration. This is consistent with the fact that siderophore-producing *Bacillus* have a better chelation activity of heavy metals, as we mentioned earlier.

The functions of *Bacillus*, such as phosphorus solubilization and potassium release, are of paramount importance for enhancing soil fertility, reducing the application of chemical fertilizers, and advancing sustainable agricultural practices [56]. Siderophore-producing bacteria (SPBs) are growth-promoting microorganisms that exhibit a high resistance to heavy metals and can alleviate the toxicity of heavy metals to plants, thereby improving plant tolerance to heavy metals [57].

We investigated the properties of WL1210 and CD303 at the physiological and genetic levels. The results show that the plant height and root length of ryegrass irrigated with the WL1210 and CD303 bacterial suspensions increased, and the chlorophyll and protein contents of ryegrass increased, while the proline content decreased, but the proline content increased with the increase in the cadmium concentration. Studies have reported a considerable proportional increase in the proline content as the concentration of heavy metals increases. Cadmium has been shown to be the strongest inducer of proline accumulation. Therefore, we hypothesized that the accumulation of proline could serve as a marker to detect the level of heavy metal pollution.

Some bacteria can adsorb heavy metal ions through functionally active groups and protein groups on the surface of their cell wall, or they can precipitate and chelate heavy metals through protein binding [58]. Siderophores can chelate cadmium ions in the soil, increasing their bioavailability and thus promoting their absorption by and translocation into plants. At the plant roots, the complexation of siderophores with cadmium ions can reduce cadmium toxicity, allowing it to enter the plant through the root system. Once within the plant, cadmium can accumulate in plant tissues, particularly in the leaves and stems [59]. This accumulation may lead to growth inhibition, physiological dysfunction, and ultimately a decrease in yield and quality [60]. Therefore, utilizing microorganisms that produce siderophores to convert cadmium in the soil into a less toxic or non-toxic state can reduce its potential harm to plants and the food chain [61].

In the process of the determination of the strains WL1210 and CD303, we found that both strains had the ability to produce iron carriers, which is very satisfactory. Subsequently, the identification of iron carrier types was carried out. The WL1210 is a catecholic type (Figure 12a), and CD303 was identified as carboxylic acid ferrophilin (Figure 12b). Carboxylate ferrophore is a siderophore-containing carboxylic acid group, which can form carboxylate complexes with cadmium ions [62], thereby chelating cadmium ions, and its chelating ability is relatively good. Through tRNA prediction after the whole-genome sequencing of CD303, it was found that it contains aspartic acid [63], which constitutes carboxylate ferrophores; so, CD303 may alleviate the contamination of the soil by cadmium ions by chelating cadmium ions through its ability to produce carboxylate ferrophores.

## 5. Conclusions

This study provides an in-depth examination of the characteristics of two *Bacillus* strains, which significantly enhance the growth of ryegrass under cadmium (Cd) stress conditions. These strains have the capability to produce siderophore molecules, which enhance plant iron absorption and thereby support plant growth. Moreover, they synthesize a variety of plant hormones, including indole acetic acid, gibberellins, and cytokinins, which play pivotal roles in signaling during plant growth and development. These hormones guide normal plant growth through synergistic or antagonistic interactions at different stages, aiding plants in adapting to environmental conditions and resisting both abiotic and biotic stresses. Additionally, these strains possess the ability to solubilize phosphorus and potassium, which is crucial for improving soil fertility and reducing the use of chemical fertilizers, thus advancing sustainable agricultural practices.

The genomic analysis further revealed the key genes within these strains that are associated with plant growth promotion and heavy metal stress response. The presence of these genes provides a scientific basis for understanding how these strains influence plant growth through molecular mechanisms. Through the functions of these genes, the strains can mitigate the toxic effects of heavy metals like cadmium on plants and enhance their tolerance in contaminated soil environments via bioremediation. This discovery not only offers new strategies for the bioremediation of soil heavy metal pollution but also provides a new perspective for improving crop growth potential under adverse conditions in agricultural production.

## Figures and Tables

**Figure 1 microorganisms-12-01083-f001:**
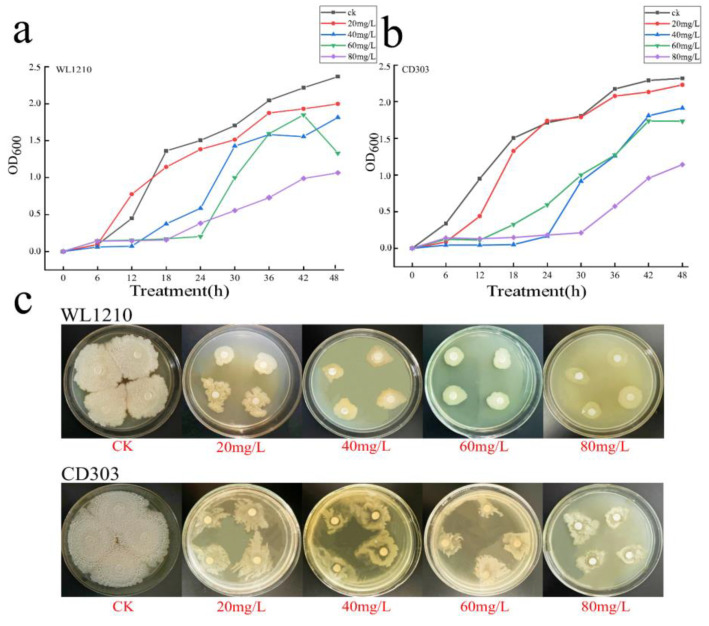
Growth of the strains under Cd^2+^ stress. Growth of the strain WL1210 under cadmium stress (**a**). Growth of the strain CD303 under cadmium stress (**b**). The growth of the two strains on a solid medium under different cadmium stresses (**c**).

**Figure 2 microorganisms-12-01083-f002:**
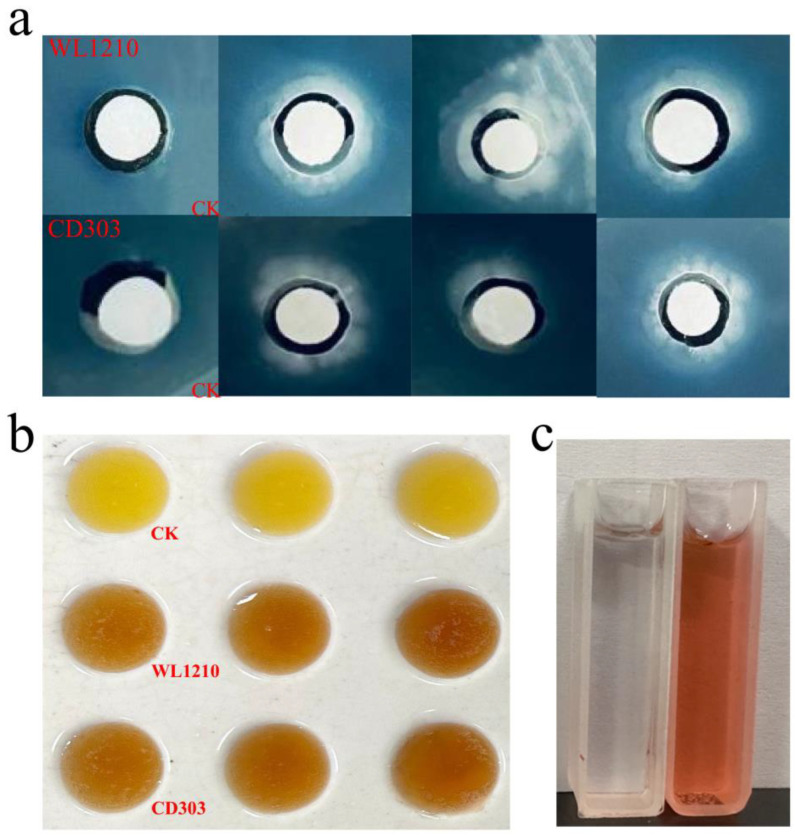
Identification of the siderophore types in the strains and determination of IAA production. Siderophore production of strains on CAS blue plates (**a**). IAA chromoproduction by strains (**b**). The color testing results of the structure types of the siderophore-chelating groups (**c**).

**Figure 3 microorganisms-12-01083-f003:**
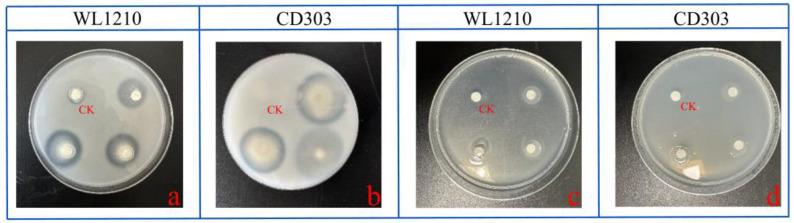
Determination of the phosphorus- and potassium-solubilizing capacities of the bacterial strains. (**a**) Phosphorus-solubilizing capacity of the strain WL1210. (**b**) Phosphorus-solubilizing capacity of the strain CD303. (**c**) Potassium-solubilizing capacity of the strain WL1210. (**d**) Potassium-solubilizing capacity of the strain CD303.

**Figure 4 microorganisms-12-01083-f004:**
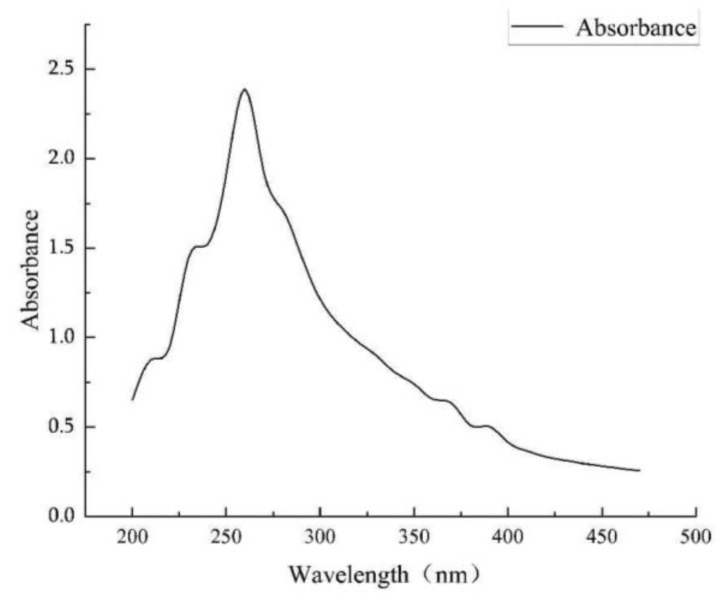
Full spectrum scan of the strain CD303’s carboxylic acid ferrophilin.

**Figure 5 microorganisms-12-01083-f005:**
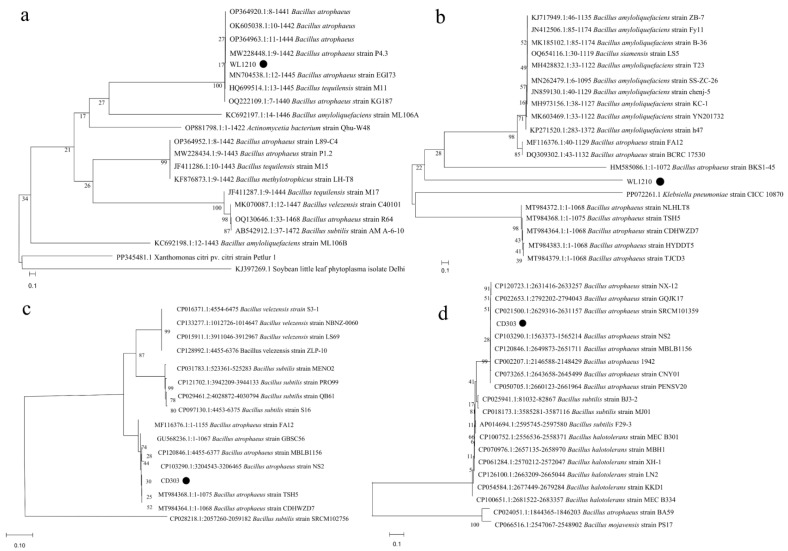
Neighbor-joining phylogenetic tree based on 16S rDNA, *gyr* B, and *dna*K partial sequences. (**a**) Phylogenetic tree constructed based on the 16S rDNA gene sequences of the strain WL1210. (**b**) Phylogenetic tree based on the *gyr* B gene sequences of the strain WL1210. (**c**) Phylogenetic tree based on the *gyr* B gene sequences of the strain CD303. (**d**) Phylogenetic tree based on the *dna*K gene sequences of the strain CD303. Bar, 0.1 substitutions per nucleotide position. Bootstrap values (expressed as percentages of 1000 replications) greater than 30% are shown at branch points. The circles in the diagram represent the strains WL1210 and CD303.

**Figure 6 microorganisms-12-01083-f006:**
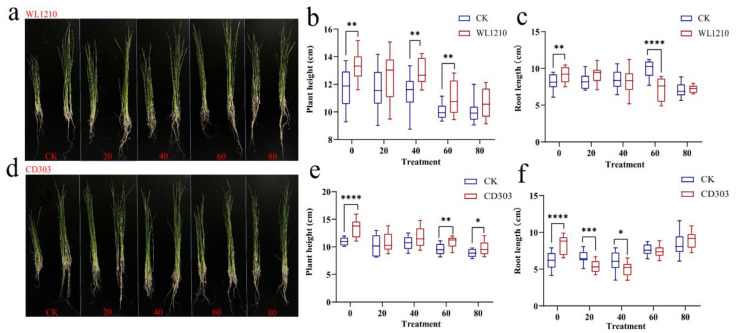
Growth promotion effect of the strain WL1210 on ryegrass (**a**). Plant height after treatment with the strain WL1210 (**b**). Root length after treatment with the strain WL1210 (**c**). Plant growth promotion effect of the strain CD303 on wheat plants (**d**). Plant height after treatment with the strain CD303 (**e**). Root length after treatment with the strain CD303 (**f**). Different asterisks at each treatment indicate significance between inoculated and uninoculated conditions at *p* ≤ 0.05 level. The * indicates a significant difference (*p* ≤ 0.05); ** indicates an extremely significant difference (*p* ≤ 0.01); *** indicates an extremely significant difference (*p* ≤ 0.001); **** indicates an extremely significant difference (*p* ≤ 0.0001) compared to the control group.

**Figure 7 microorganisms-12-01083-f007:**
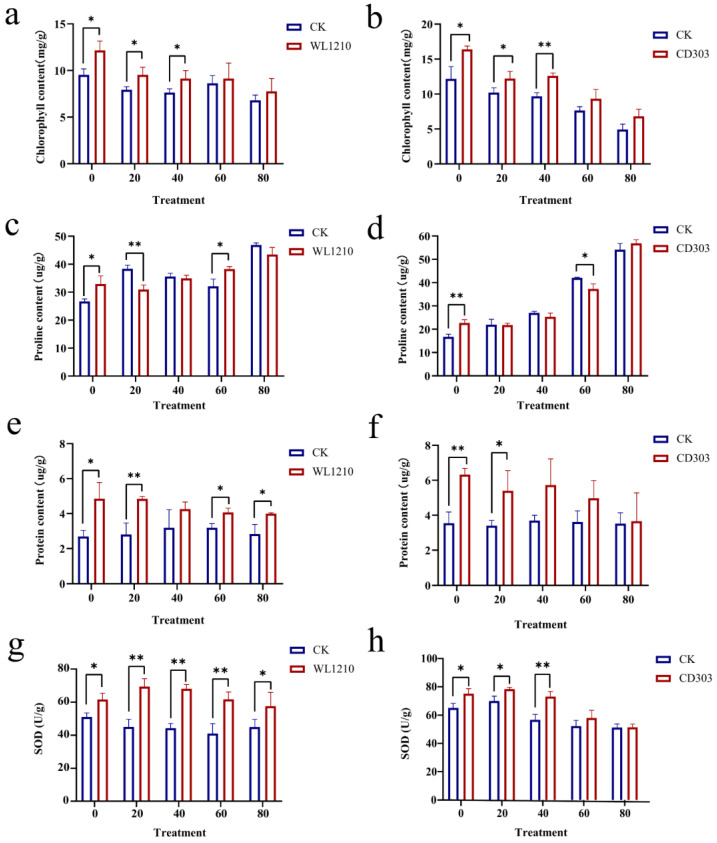
Ryegrass chlorophyll contents after Cd^2+^ exposure and treatment with the strains (**a**,**b**). Ryegrass proline content after Cd^2+^ exposure and treatment with the strains (**c**,**d**). Ryegrass protein concentrations after Cd^2+^ exposure and treatment with the strains (**e**,**f**). Ryegrass superoxide dismutase concentrations after Cd^2+^ exposure and treatment with the strains (**g**,**h**). Different asterisks at each treatment indicate significance between inoculated and uninoculated conditions at *p* ≤ 0.05 level. The * indicates a significant difference (*p* ≤ 0.05); ** indicates an extremely significant difference (*p* ≤ 0.01).

**Figure 8 microorganisms-12-01083-f008:**
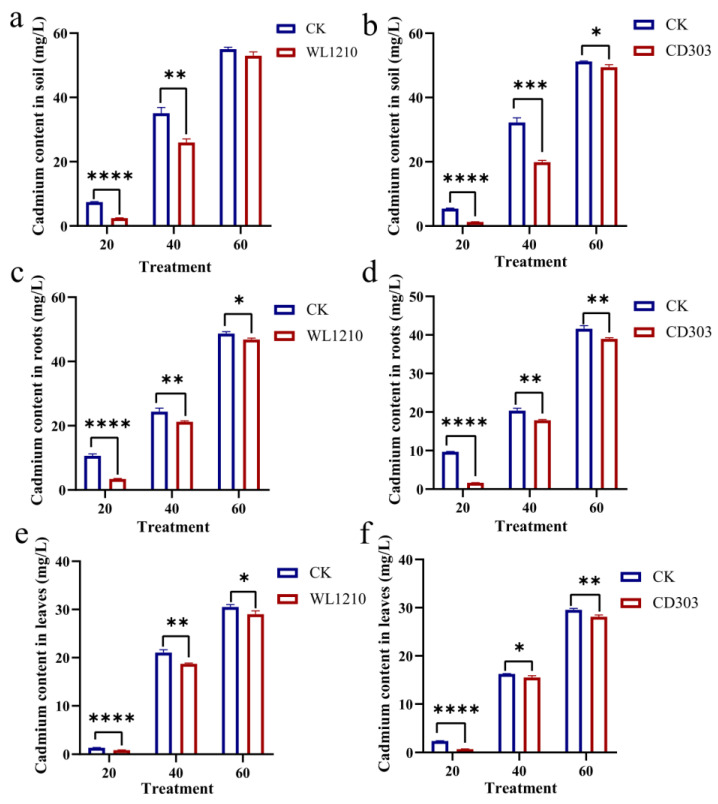
Determination of the cadmium content in soil and plants. Cadmium content in soil under different treatments (**a**,**b**). Cadmium content in ryegrass roots under different treatments (**c**,**d**). Cadmium content in ryegrass leaves under different treatments (**e**,**f**). Different asterisks at each treatment indicate significance between inoculated and uninoculated conditions at *p* ≤ 0.05 level. The * indicates a significant difference (*p* ≤ 0.05); ** indicates an extremely significant difference (*p* ≤ 0.01); *** indicates an extremely significant difference (*p* ≤ 0.001); **** indicates an extremely significant difference (*p* ≤ 0.0001) compared to the control group.

**Figure 9 microorganisms-12-01083-f009:**
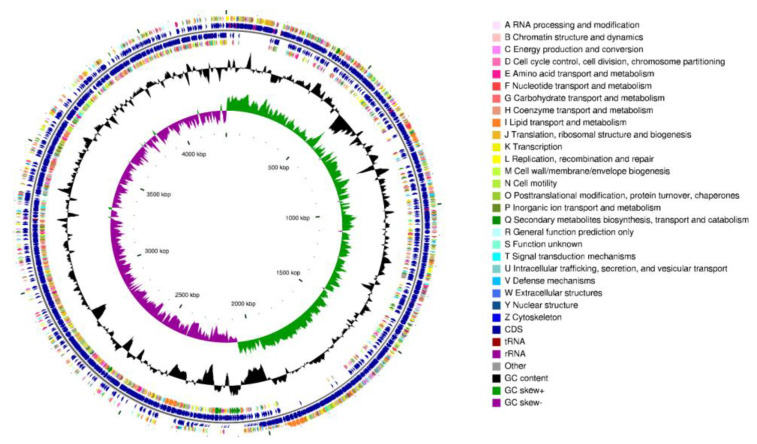
Circular map of the *Bacillus* CD303 genome.

**Figure 10 microorganisms-12-01083-f010:**
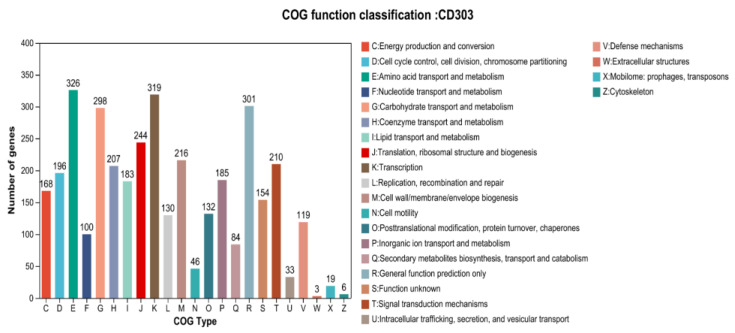
Functional annotation of the *Bacillus* CD303 strain.

**Figure 11 microorganisms-12-01083-f011:**
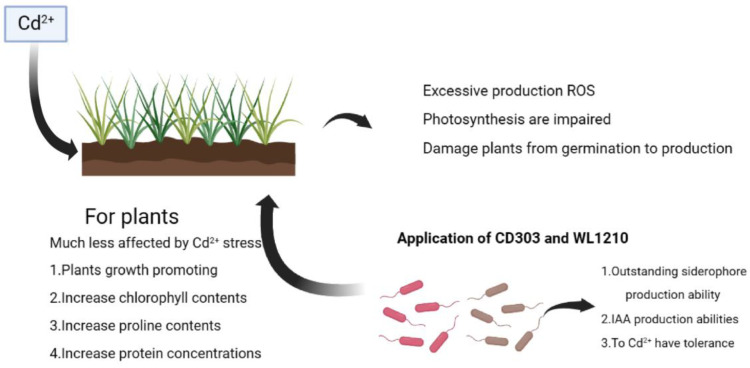
Pathways of *Bacillus* CD303 and WL1210 to alleviate ryegrass under heavy metal stress.

**Figure 12 microorganisms-12-01083-f012:**
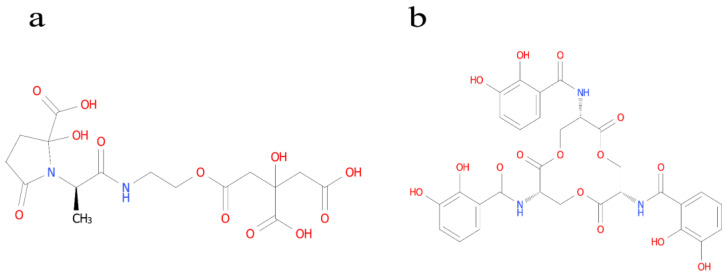
Structure of the catecholate siderophore enterobactin (**a**) and the carboxylate siderophore vibrioferrin (**b**).

**Table 1 microorganisms-12-01083-t001:** Siderophore production activity of the strains.

Strain	Siderophore Activity (%)	A/Ar
WL1210	36.16	++
CD303	49.88	+++

**Table 2 microorganisms-12-01083-t002:** Effect of different Cd^2+^ concentrations on the seed germination rates of ryegrass.

Treatment Concentration (mg/L)	Germination Rate per Day after Bed Placement (%)			
	7 d	8 d	9 d	10 d
0	78.0 ± 4.00	81.3 ± 5.77	85.3 ± 9.45	91.3 ± 6.11
20	76.0 ± 5.29	78.6 ± 8.08	84.0 ± 12.00	88.0 ± 12.41
40	78.0 ± 3.46	83.3 ± 6.42	84.0 ± 9.16	88.7 ± 9.45
60	77.3 ± 6.11	81.3 ± 5.77	83.0 ± 8.71	83.3 ± 9.45
80	18.0 ± 4.00	18.6 ± 3.06	29.3 ± 11.37	34.7 ± 11.79

**Table 3 microorganisms-12-01083-t003:** Effects of different Cd^2+^ concentrations on the germination index (GI) and viability index (VI) of ryegrass seeds.

Concentration (mg/L)	GV (%)	GR (%)	GI	VI
0	78.0 a	91.3 a	5.0 ± 0.44 a	15.9 ± 3.04 a
20	76.0 a	88.0 a	4.9 ± 0.44 ab	13.5 ± 1.09 ab
40	78.0 a	88.7 a	5.0 ± 0.50 a	10.9 ± 1.06 bc
60	77.3 a	83.3 a	4.9 ± 0.69 a	8.7 ± 1.14 c
80	18.0 b	34.7 b	1.5 ± 0.27 b	1.4 ± 0.59 d

**Table 4 microorganisms-12-01083-t004:** Relevant functional genes identified in the genome of CD303.

Gene Name	Location	Length	Function
*glpQ*	236,068–236,355	288 bp	Glycerophosphodiester phosphodiesterase
*hxlA*	380,639–381,271	633 bp	3-hexulose-6-phosphate synthase
*hxlB*	380,075–380,632	558 bp	Carbohydrate derivative metabolic process, isomerase activity, and carbohydrate derivative binding
*miaA*	1,957,705–1,958,649	945 bp	Isopentenyl adenine gene, considered the main cytokinin
*miaB*	1,861,144–1,862,673	1530 bp	Isopentenyl adenine gene, considered the main cytokinin
*ndhF*	197,749–199,266	1518 bp	NADH dehydrogenase subunit 5, chloroplast functional genes
*YBCL*	204,425–205,597	1173 bp	MFS transporter, encodes the Rubisco enzymes in plants, and is capable of converting carbon dioxide into organic matter
*frmA*	372,000–373,136	1137 bp	Threonine dehydrogenase or related Zn-dependent dehydrogenase
*phnB*	408,798–409,163	366 bp	Zn-dependent glyoxalase, PhnB family
*fnr*	3,382,739–3,383,740	1002 bp	Ferredoxin and NADP reductase 2
*iscA*	3,385,318–3,385,680	363 bp	Iron-sulfur cluster assembly protein
*ggt*	2,146,580–2,148,346	1767 bp	Plant stress tolerance enhancer
*mgtC*	791,225–791,917	693 bp	Magnesium uptake protein YhiD/SapB, involved in acid resistance
*proB*	2,158,357–2,159,481	1125 bp	Involved in proline or betaine synthesis and alleviates cell apoptosis
*proC*	2,159,505–2,160,305	801 bp	Involved in proline or betaine synthesis and alleviates cell apoptosis
*glnA*	1,969,335–1,970,669	1335 bp	Involved in the regulation of nitrogen fixation
*glnR*	1,968,870–1,969,277	408 bp	Involved in the regulation of nitrogen fixation
*licA*	382,536–393,302	10,767 bp	Surfactin non-ribosomal peptide synthetase SrfAA
*licB*	393,324–404,093	10,770 bp	Surfactin non-ribosomal peptide synthetase SrfAB
*licC*	404,112–407,942	3831 bp	Non-ribosomal peptide synthetase
*srfATE*	407,970–408,704	735 bp	Surfactin biosynthesis thioesterase SrfAD
*acpT*	411,550–412,224	675 bp	4′-phosphopantetheinyl transferase superfamily protein
*tcyC*	413,260–414,003	744 bp	Amino acid ABC transporter ATP-binding protein

## Data Availability

Data are contained within the article.

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
