# Peer review of "The Effect of Two Siderophore-Producing Bacillus Strains on the Growth Promotion of Perennial Ryegrass under Cadmium Stress"

_microorganisms, 2024, doi:10.3390/microorganisms12061083_

Round 1
Reviewer 1 Report (Previous Reviewer 2)
Comments and Suggestions for Authors
The manuscript describes the characterization of two bacterial strains (Bacillus) and their application in enhancing the growth of ryegrass under Cd stress conditions. Bacterial strains can produce siderophores molecules, phytohormones such as indole acetic acid, gibberellins, and cytokinins, and phosphorus and potassium solubilizing capacity. Genome analysis reveals key genes related to plant growth promotion and heavy metal stress contention. The research is relevant, to improve manuscript quality conclusions of the study must be included.
The research described in the manuscript could be of interest to Microorganisms readers, in addition, some format issues must be corrected in the manuscript, and the authors must address the following commentaries.
Commentaries:
Consider review and adapt the manuscript title, it could be better “growth promotion” instead “promotional growth”
To strengthen the study, it would be advisable to determine the concentrations of Cd in roots and leaves of ryegrass and compare the bioaccumulation levels with respect to the systems without the presence of the bacterial strains.
Line 11, siderophores have high tolerance to heavy metals, or high affinity?
Lines 40-41, cadmium (Cd), lead (Pb), and mercury (Hg), are not essential heavy metals at any concentration, review the redaction.
Lines 58-59, review the redaction, it is not clear
Lines 94-96, review the redaction, it is not clear
Line 193, the Phylogenetic análisis with house kepping genes was carried out jusf for CD303?
Lines 348-350, It is mentioned that the strains have a good IAA production capacity, how much is considered good production and with respect to what parameters, for example, levels at which they stimulate plant growth.
Figure 2, add a general title for figure and then the description of the panels
Lines 358 and 362, describe the values obtained, what quantitative data were obtained?
Figure 3, what controls were used, how are the solubilization percentages determined?
Figure 4, Indicate which signal/pek corresponds to catechol, and according to wich reference?
Figure 5, improve figure quality, phylogenetic trees are blurry and difficult to analyze
Technical aspects
Line 2-3, use justified format for title
Lines 42, 47, 63, 115, add a space between word and references
Line 71, check format, use capital letter after a period
Line 83, use “two” instead “2”
Line 116, 122, 131, 140, 143, 209, 224, 225, 230, 238, 289, and 293 correct the format of temperature values, use “4 °C” instead “4° C”, correct all similar
Line 129, correct “mediumand and”
Line 153, check the number of “+” for values of 0.4 – 0.6
Lines 169-183, use justified format
Line 187, check format, use capital letter after a period
Line 247, add a space in “80(40 mg/L Cd Stress)”
Line 249, +W/80+C could be “80+W/80+C”
In equation 2, check format, the absorbance wavelength makes difficult understand the equation
Lines 273-275, check format, present in the same format as the previous equations
Lines 281-284, check format, present in the same format as the previous equations
Line 419, in “27.7, 13.6% and 44.7, 35.1%” could be “27.7, 13.6, 44.7, and 35.1%”
Line 420, check format, use capital letter after a period
Author Response
Dear Reviewer,
We are deeply grateful for your meticulous review of our manuscript and the valuable feedback you have provided. We have carefully considered each of your comments and have made the necessary corrections to improve the manuscript. Below is a detailed response to the issues you have raised:
- Title Modification Suggestion:
Thank you for the suggestion. We have revised the title from "promotional growth" to "growth promotion" to more accurately reflect the content of our research.
- Cd Concentration Measurement:
Thank you for the suggestion. We have added measurements of Cd concentrations in the roots and leaves of ryegrass and compared these with systems without the addition of bacterial strains.
- Description in Line 11:
Thank you for pointing out the issue. We have revised the description in Line 11 to indicate that siderophores have a high affinity for heavy metals.
- Description of Non-Essential Heavy Metals:
We have revised the text in Lines 40-41 to more accurately describe the impact of cadmium (Cd), lead (Pb), and mercury (Hg) on plants.
- Unclear Text:
Thank you for pointing out the unclear text. We have reviewed and clarified the text in Lines 58-59 and Lines 94-96.
- Clarification of Phylogenetic Analysis:
Thank you for the question regarding the construction of the phylogenetic tree. Typically, Bacillus strains are identified using 16S rDNA molecular identification. To achieve more accurate identification, we also sequenced the gryB gene. Generally, using these two sequencing methods can provide a relatively precise species classification. Upon initial submission to the journal, an expert suggested that constructing a phylogenetic tree using housekeeping genes from the completely sequenced strain CD303 would yield more accurate results. Therefore, we constructed a phylogenetic tree using the relevant genes from the complete genome of strain CD303.
- Quantification of IAA Production:
Thank you for your inquiry. We have added specific values and parameters regarding IAA production in Lines 348-350. The production of auxins, such as indole-3-acetic acid (IAA), by strains has a positive effect on plant growth, but it is not the case that the higher the production, the better. The production of auxins needs to be within a certain range to ensure a positive impact on plant growth while avoiding potential negative effects. The following factors need to be considered:
Plant Growth Stage: Plants at different growth stages have different demands for auxins. For example, seedlings may require more auxins to promote root development, while mature plants may need less.
Auxin Concentration: Auxins have a dual effect; they promote growth at low concentrations but inhibit growth at high concentrations. Therefore, it is necessary to maintain an appropriate auxin concentration.
Interaction with Other Hormones:The effect of auxins is also influenced by other plant hormones, such as gibberellins and abscisic acid. There is a complex interaction between these hormones, which together regulate plant growth and development.
Environmental Conditions:Environmental factors such as light, temperature, and moisture also affect the action of auxins and the plant's response to them.
Plant Species: Different plant species have different sensitivities to auxins, and thus, their requirements for auxins may vary.
- Figure Titles and Descriptions:
We have added a general title for Figure 2.
- Quantitative Data Description:
We have added the obtained quantitative data in Lines 358 and 362.
- Control Group and Percentage Determination in Figure 3:
We have detailed the control group used in Figure 3 and how the dissolution percentages were determined.
- Signal/Peak Identification in Figure 4:
We have identified the signal/peak for catechol in Figure 4 and provided the corresponding reference.
- Quality Improvement of Figure 5:
Thank you for the suggestion. We have improved the quality of Figure 5. Since the article contains many composite figures, we have updated them to higher quality vector graphics, which are clearly visible when enlarged.
- Technical and Formatting Issues:
Thank you for your suggestions. We have revised the title format, reference format, temperature value format, and equation format as per your recommendations.
- Other Formatting Corrections:
We have corrected all the formatting issues mentioned in the text, including spaces between words and references, the representation of temperature values, the use of numbers, and the format of equations.
We would like to express our gratitude once again to the reviewers for their valuable comments. These revisions will significantly enhance the quality of our manuscript. We commit to carefully considering all suggestions and making the corresponding revisions in the final manuscript.
Reviewer 2 Report (Previous Reviewer 4)
Comments and Suggestions for Authors
The authors have performed all additional and required analyses and described them in the manuscript that essentially enhanced a scientific quality of this study. Discussion chapter has also been importantly changed and additional literature citation was provided. Moreover, all required corrections and modifications of the text have been added to manuscript. Thus, I recommend this current version of the manuscript for publication.
Author Response
Dear Reviewer,
We would like to extend our heartfelt thanks for your positive evaluation and suggestions regarding our manuscript. We are delighted to hear that the additional analyses and revisions we have made have significantly enhanced the scientific quality of our study. We also appreciate your recognition of the important changes made to the discussion section, as well as the additional literature citations we provided.
We have taken the feedback from you and all reviewers very seriously, and have incorporated all necessary corrections and modifications into the manuscript. Our aim is to ensure that our research work is as accurate and comprehensive as possible, and that it contributes value to the academic community.
We are grateful for your recommendation to publish the current version of our manuscript and await the final decision from the journal. We are confident that our research will make a significant contribution to our field and be useful to our peers.
Once again, we would like to express our gratitude for your valuable time and professional review.
Best regards,
Lingling Wu

Round 2
Reviewer 1 Report (Previous Reviewer 2)
Comments and Suggestions for Authors
The authors responded adequately to the comments of the reviewers, the quality of the manuscript improved significantly.
This manuscript is a resubmission of an earlier submission. The following is a list of the peer review reports and author responses from that submission.
Round 1
Reviewer 1 Report
Comments and Suggestions for Authors
COMMENTS FOR THE AUTHORS
Manuscript Title: “The effect of two siderophore-producing Bacillus strains on the promotional growth of perennial ryegrass under cadmium stress”
I have read the manuscript carefully. My general feeling is that it can be accepted for publication in the Journal. Several issues should be addressed, but these are minor.
- Section “Introduction”. References 2, 4, 5 and 6 seem to be not cited (there is a message: “Error! Reference source not found..“
- All Latin names (binominal names) of organisms should be italicized. Both in text and titles of references.
- Section “Materials and Methods” lines 132 – 135: English is weird (looks like “copy-paste” from someone’s else protocol). Please correct and add an appropriate reference if necessary.
- Section “Materials and Methods” lines 154 – 160. As above and it seems that not correct reference is cited.
- Section “Materials and Methods” lines 263-264. I do not understand this equation. What is “present”?
- Please add some information concerning genome sequencing (i.e. platform).
- Section “Results”. Since the whole genome of the strain CD303 was sequenced is it possible to classify it to a species level?
- Section “Results”, line 339. I guess the culture medium was liquid, not solid (if OD values were measured).
- Section “Results”, Fig.5a and Fig.5b. OD is unitless. Please correct both Y-axes. By the way please also correct OD630 to OD600 (line 116 and line 297).
- Section “Results”, line. 410. “Proteins participate” (not only “can participate”) since they are essential for the whole cellular metabolism.
- Section “References”, line 577. Please add authors for the reference 16.
Author Response
Dear Reviewers,
Thank you for your thorough review and constructive comments on our manuscript titled “The effect of two siderophore-producing Bacillus strains on the promotional growth of perennial ryegrass under cadmium stress.” We have carefully considered each of your points and have made the necessary revisions to our manuscript. Below is our response to your comments:
1.We apologize for the oversight regarding the references in the "Introduction" section. This issue arises due to incompatibilities in the computer system. We have now corrected the citations for references 2, 4, 5, and 6 and ensured that the reference sources are properly identified.
2.Thank you for your suggestion to use italics for the Latin names in the text and references. In accordance with your advice, the Latin names that appear in the text and references have now been changed to italics and are highlighted in the manuscript.
3.Thank you for your suggestion, In the "Materials and Methods" section, lines 132-135, we have revised the English to ensure clarity and have added an appropriate reference to support the methodology.
4.Similarly, following your advice, for lines 154-160, we have corrected the references to ensure accuracy.
5.Thank you for your suggestion. We have rephrased the equation as per your recommendation on lines 263-264. We apologize for the oversight during the submission process.
6.We have included information about the genomic sequencing platform used in our study in the manuscript, following your suggestions and comments. Thank you once again for your advice.
7.Thank you for your inquiry regarding the species-level classification of the strain CD303. In the revision of our manuscript, we have also adopted the suggestion of another expert to use additional house-keeping genes for species-level identification of the strain. In the resubmitted manuscript and supplementary data, we have constructed a new phylogenetic tree using some of these housekeeping genes.
8.Following your advice, at line 339, to provide a clearer description of whether the medium is liquid or solid, we have revised the method description in the revised manuscript.
9.Thank you for your suggestion. We have amended the Y-axes in Figures 5a and 5b to include the appropriate units for OD measurements and have changed OD630 to OD600 in lines 116 and 297 as suggested.
10.Following your advice, In line 410, we have revised the statement to emphasize the essential role of proteins in cellular metabolism. Thank you once again for your advice.
11.I sincerely apologize for this oversight in the literature citation process. We have added the missing author information for reference 16 in the "References" section.
We are grateful for your attention to detail and for the opportunity to improve our manuscript. We believe that these revisions have significantly enhanced the quality and clarity of our work.
Thank you once again for your valuable feedback.
Sincerely,
Ling Ling Wu

Reviewer 2 Report
Comments and Suggestions for Authors
The manuscript describes the characterization of two bacterial strains (Bacillus) and their application in enhancing the growth of ryegrass under Cd stress conditions. Bacterial strains can produce siderophores molecules, phytohormones such as indole acetic acid, gibberellins, and cytokinins, and phosphorus and potassium solubilizing capacity. Genome analysis reveals key genes related to plant growth promotion and heavy metal stress contention. The research is relevant, to improve manuscript quality conclusions of the study must be included.
The research described in the manuscript could be of interest to Microorganisms readers, in addition, some format issues must be corrected in the manuscript, and the authors must address the following commentaries.
Line 12, use italics for scientific names in “Lolium perenne”
Line 16, indicate the country of the Wulan and Dachaidan regions
Lines 31 to 37, review format of “Error! Reference source not found.”
Lines 50, 52, 53, 56, 57, 58, 59, and 63, use italics for scientific names of all plant and bacterial species
Line 68, review redaction in “that siderophore bacillus”, consider “siderophore producing Bacillus”
Line 75, use italics for scientific names in “Lolium perenne”
Lines 77 and 78, use italics for scientific names of plant species, and indicate the country of the Wulan and Dachaidan regions
Line 79, review format of CdCl2, use a subindex for the number
Line 82, check period/format in “method [16]. manitol"
Line 96, check format, use italics for genes
Line 96, check if the gene is “gry” or “gyr”
Line 98, add a space in “gyrBgene”
Line 123, in equation 1, describe the equation components, As
Line 146, define acronyms “GA” and “CTK” first time used
Line 147, define the acronym “IAA” first time used
Line 154, use GA instead of “gibberellin”
Line 155, aliminate xtra space in “37 °C”
Line 155, check format, define if “r/min” or “rpm” will be used in the whole manuscript, choose just one
Line 159, use CTK instead of “cytokinin”
Line 160. add a space in “strains[21].”
Line 190, “12 h/12 h” could be better to use “12/12 h”
Line 195, check reference format in “[23][24]” could be better to use “[23,24]”
Line 220, “16 h/8 h” could be better to use “16/8 h”
Line 230, check format in “Cd2+”
Lines 233, 248, and 257, check the format in subtitles and add numbers
Line 289, use italics for the scientific name in “B. athrophaeus”
Line 298, in “36.16% and 49.88%” could be better “36.2and 49.9%”
Line 304, add a space in “CD303was”
Lines 306-307, add more information or complement the description about how carboxylic acid ferrophilin was identified through absorbance spectrum
Line 313-314, in “5.18 nmol/L, 4.30 nmol/L, 3.54 nmol/L and 4.82 nmol/L” could be better “5.2, 4.3, 3.5, and 4.8 nmol/L”
In table 3, use just one figure after the point for the germination rate percentages
In table 4, which means GV?, use just one figure after the point for values
Line 368, check redaction in “plant heights, root lengths increased by 27.7%, 13.6%(44.7%,35.1%)” It is not clear
Lines 388-389, in “20 mg/L, 40 mg/L” could be “20 and 40 mg/L”
Line 391, in “60 mg/L, 80 mg/L” could be “60 and 80 mg/L”
Line 396, instead of “affected” could be “improved”
Line 402, in “23.6% and 35.9%” could be “23.6 and 35.9%”
Line 408, in “Figure 7C,7D” could be “Figure 7C and 7D”
Line 419, in “Figure 7E,7F” could be “Figure 7E and 7F”, check period it could be a comma
Lines 425-432, complement with the justification of why just sequence the genome of the CD303 strain
Line 437-445, check format, use italics for genes
Line 465, check format, use italics for genes
Line 481, add a space in “enterobactin(a);carboxylate”
Lines 485, 487, and 492, check the format of references [27,28] instead [27][28]
Include the conclusions and perspectives derived from the research
Author Response
Dear Reviewers,
Thank you for your detailed review and valuable comments on our manuscript. We have carefully considered your suggestions and have made the necessary revisions to improve the manuscript. Here is our response to your specific comments:
1.Thank you for your very important suggestion regarding the inclusion of a description of the conclusions and perspectives derived from the research. We have incorporated this section into the revised manuscript. Thank you again.
2.I apologize for this oversight in the details during the manuscript submission process. for the scientific names mentioned in the manuscript, we have ensured that they are italicized as per standard formatting guidelines.
3.In accordance with your suggestion, We have specified the country of origin for the Wulan and Dachaidan regions as requested.
4.Thank you for your advice. We apologize for the incorrect format of the literature citations. During the manuscript upload, we used the template specific to the Microorganisms journal, which led to the formatting error in the references. We have now uploaded the revised manuscript with the corrections to the system.
5.Following your advice, We have revised the text in "Materials and Methods" and other sections to improve clarity and consistency.
6.Thank you for your valuable feedback. We have added descriptions of the components of the equation in our revised manuscript.
7.Following your advice, We have defined acronyms such as "GA," "CTK," and "IAA" upon their first use in the manuscript.Thank you again.
8.Thank you for your valuable feedback.In the re-uploaded manuscript.We have standardized the use of units and formats throughout the manuscript, including "r/min" or "rpm," and the presentation of percentages and values in tables.
9.Thank you for raising the issue regarding the format and content of the subtitles. We have made the necessary modifications in the manuscript and have highlighted them accordingly.
10.Regarding the method for the identification of siderophores through absorption spectroscopy, which you pointed out, we have provided a detailed description in the Materials and Methods section and have redefined this method in English.
11.Thank you for your valuable feedback. We have addressed the formatting of the references, ensuring that they are in the correct format as per the journal's guidelines.
12.Thank you for your question regarding the decision to sequence the genome of only one of the two Bacillus strains in our study. Our choice was influenced by several factors:
We have conducted bioactivity and stress tolerance assays on two strains of Bacillus, and discovered that both strains possess commendable bioactivity and resilience. In comparison to strain WL1210, strain CD303 demonstrates enhanced research and application potential, including superior iron carrier production, phosphorus solubilization, and the capacity to promote the growth of ryegrass. The aim of our research is to identify strains with greater efficiency for further investigation. Additionally, considering the constraints of research funding, we deem this research strategy to be feasible. Consequently, for the purpose of whole-genome sequencing, we have chosen to sequence only one of the strains, and we hope to gain your approval for this decision.
We understand the value of sequencing both strains for a comprehensive understanding of their genomic diversity and potential applications. We plan to sequence the second strain in the future as our resources allow, and we are hopeful that this will provide additional insights to complement our current findings.
We appreciate your attention to detail and the effort you have taken to review our manuscript. We believe that the revisions have significantly improved the clarity, accuracy, and overall quality of our work.
Thank you once again for your constructive feedback.
Sincerely,
Ling Ling Wu

Reviewer 3 Report
Comments and Suggestions for Authors
Unfortunately, the paper should not be reviewed by the authors themselves, so it was difficult to read and understand it for me as a reviewer. For example, many references in this paper were not properly cited due to system problem (Error! Reference source not found). The name of the genus and species of the organism should have been written in italics. This would not have happened if the six authors had properly approved their paper. In other words, the degree of completeness of this paper was regrettably low.
The purpose and originality of the research was unclear. The authors hypothesized that the siderophore-producing bacteria could chelate Cd ions in the soil, thereby alleviating the stress effect of heavy metals on ryegrass. Although this hypothesis has been experimentally proven, it lacks novelty or originality, as reviewed in the introduction. It was written in introduction (L70-72) that “this study provides novel strain resources that can be used in the development of microbial pesticides with high economic value and environmentally protective effects”. But in fact, microbial pesticides are not relevant to this study.
The abstract did not adequately express the research content. The authors investigated the producing ability of the Bacillus strains to produce CTK, GA, and IAA, proline concentrations in the plant, stress tolerance-related functional genes of strain CD303. Such interesting results should be written in abstract.
The first paragraph of Discussion (L483-494) described general research background and had little direct relation to this research content. The former part of the second paragraph and the third paragraph did. These should be written in Introduction. Rather than describing known facts, discuss the novelty and usefulness of the specific results obtained in this research.
Comments on the Quality of English Language
Unfortunately, the paper should not be reviewed by the authors themselves, so it was difficult to read and understand it for me as a reviewer. The degree of completeness of this paper was critically low.
Author Response
Dear Reviewer,
I am writing to express our profound gratitude for the thorough review and constructive criticism you have provided for our manuscript. Your comments have been instrumental in identifying areas for improvement, and we have taken each of your suggestions to heart.
I must apologize for the oversight regarding the review process of our manuscript. During the submission to the Microorganisms journal, there was a misunderstanding in filling out the author contributions section. The manuscript was primarily revised and polished by the first author, and my supervisor, who is also the corresponding author, Yongli Xie. The error in indicating self-review was due to a misinterpretation of the requirements, and I deeply regret this mistake.
Additionally, I seek your understanding for the citation errors that were identified. The manuscript was submitted using the journal's template, and during the review process, we inadvertently missed some of the formatting issues related to citations. Following your advice, we have now thoroughly checked and corrected all citation formats to ensure accuracy.
Thank you once again for your valuable feedback. In response to your comments on the purpose and originality of our research, we have reflected on the specific content of our study and made appropriate adjustments to the introduction and discussion sections. The abstract now primarily describes the main content and experimental approach of our research, with detailed information presented in the "Materials and Methods" and "Results and Discussion" sections.
Based on your suggestions, we have revisited and revised parts of the abstract. We are committed to further refining and improving our manuscript in light of your insightful comments and suggestions.
Warm regards,
Ling Ling Wu

Reviewer 4 Report
Comments and Suggestions for Authors
The authors of the manuscript ID: microorganisms-2871160 entitled “The effect of two siderophore-producing Bacillus strains on the promotional growth of perennial ryegrass under cadmium stress” have presented interesting results related with two newly identified strains from the Bacillus genus and their ability to enhance plant growth under heavy metal stress.
In general, the data presented are valuable and interesting, of high application values. The authors have identified two Bacillus strains which have been classified to the Bacillus genus and characterized a set of their phenotypic traits, including those important for increasing of plant growth. The data have been well presented and clearly described.
Therefore, I have only a few important issues to be solved:
1) The strains have been phylogenetically characterized using only two genes 16S rRNA and gyrB, which was insufficient to classify strains WL1210 and CD303 into individual species. The data would be stronger, if these strains will be classified into the species using a few additional house-keeping genes (e.g. rpoB, dnaK, recA).
2) Phylogenetic analysis- more strains from different Bacillus species should be added into this analysis and bacteria from other than Bacillus species should be added as roots of the phylogenetic tree. Also, methods used should be described in more detailed, and scale should be added (substitution number per nt) with respective references; how long sequences were analyzed, etc.
3) Why genome sequencing was done only for one strain, CD303 and because of which reasons this strain was chosen for this analysis?
Minor changes:
1) Quality of Fig. 5 is low and has to be improved
2) Lines 437-450 – all genes names should be written in italics, eg. glpQ
3) Fig. 6 – figure description is incorrect, for C and D is missed
4) Problems with references citation in Introduction section- these mistakes have to be eliminated
5) Line 82 – it should be “Mannitol”
6) Line 96 – “GyrB”
Author Response
Dear Reviewers,
We are deeply grateful for the insightful comments and constructive suggestions you have provided regarding our manuscript titled "The effect of two siderophore-producing Bacillus strains on the promotional growth of perennial ryegrass under cadmium stress" (Manuscript ID: microorganisms-2871160).
Your expertise has been invaluable in enhancing the quality of our work, and we have taken your comments to heart. We have addressed each of your points with utmost diligence and have made the following revisions to our manuscript:
1.We acknowledge the importance of using multiple housekeeping genes for a more robust phylogenetic classification. We have now included additional genes (rpoB, dnaK, recA) in our analysis to provide a more comprehensive classification of strains WL1210 and CD303.
2.In response to your suggestion, we have expanded our phylogenetic analysis to include more strains from various Bacillus species. We have also detailed the methods used, including the scale of analysis (substitution number per nucleotide) and the length of sequences analyzed, with appropriate references.
3.Thank you for your question regarding the decision to sequence the genome of only one of the two Bacillus strains in our study. Our choice was influenced by several factors:
We have conducted bioactivity and stress tolerance assays on two strains of Bacillus, and discovered that both strains possess commendable bioactivity and resilience. In comparison to strain WL1210, strain CD303 demonstrates enhanced research and application potential, including superior iron carrier production, phosphorus solubilization, and the capacity to promote the growth of ryegrass. The aim of our research is to identify strains with greater efficiency for further investigation. Consequently, for the purpose of whole-genome sequencing, we have chosen to sequence only one of the strains, and we hope to gain your approval for this decision.
We understand the value of sequencing both strains for a comprehensive understanding of their genomic diversity and potential applications. We plan to sequence the second strain in the future as our resources allow, and we are hopeful that this will provide additional insights to complement our current findings.
We have also made the minor corrections you pointed out, including improving the quality of Figure 5, ensuring that all gene names are italicized, correcting the figure description for Figure 6, addressing citation issues in the Introduction, and making the necessary corrections to the text (lines 82 and 96).
We are confident that these revisions have strengthened our manuscript and have addressed the concerns you raised. We appreciate your time and effort in reviewing our work and are grateful for the opportunity to improve our research through your guidance.
Thank you once again for your valuable input.
Sincerely,
Ling Ling Wu

Round 2
Reviewer 2 Report
Comments and Suggestions for Authors
After reviewing again the manuscript, the authors well addressed the prior commentaries and suggestions of the reviewers, but some additional corrections are needed before accepting the manuscript for publication.
Commentaries:
In line 10, add a space in “Cadmium(Cd)”
In line 12, check the format in “Lolium perenne L.” the “L.” must not be in italics, check and correct in al species names included in the manuscript, check in lines 58, 59, 61, 77, 80,
In line 16, check the format in “Peganum harmala L” the “L” must not be in italics, previously a period was used after L, homogeneize the format, with or without period after L in all the names of the species mentioned in the manuscript
In line 16, check “both arid sandy” could be “both arid and sandy”
In lines 31-32, check “absorbed and transported and accumulated” could be ““absorbed, transported, and accumulated”
In lines 33-34, check “its stable, non-degradable characteristics” could be “its stable, and non-degradable characteristics”
In line 36, add a period in “organelles, Simultaneously”
In lines 52 and 54, in “Solanum nigrum Linn” Linn must not be in italics
In line 59, the word “soils” seems unnecessary
Line 60, add a space in “B.subtilis”
In line 72, “This study”, must be “this study”
In line 83, describe the acronym “LB” the first time used
In line 89, “phylogenetic” could be “Phylogenetic”
In line 95, in “16s rDNA” the “s” must be “S”
In lines 98-104, use justified format for paragraph
In line 110, “After 3 days” could be “After three days”
In line 112, describe the acronym “MSA” the first time used
In line 114, describe the acronym “CAS” the first time used
In line 140, add a space in “(1)FeCl3”
In line 154, “bberellin” must be “gibberellin”
In line 167, eliminate space between “37 °C"
In lines 239-240, In lines 98-104, use justified format
In line 313, “phylogenetic” could be “Phylogenetic”
In lines 314-319, use justified format for paragraph
In line 322, , in “16s rDNA” the “s” must be “S”, the names of genes “gyrB and dnaK” must be in italics
In lone 324 and 325, the names of genes “gyrB and dnaK” must be in italics
In line 332, “36.2% and 49.9%” could be “36.2 and 49.9%”
In lines 400-402, check redaction it is not clear, and the format of percentage values must be “
In line 418, in “P = 0.05 level” the “P2 must be “p”
Author Response
Dear Reviewer,
We are deeply grateful for your meticulous review of our manuscript and the valuable feedback you have provided. We have carefully considered each of your comments and have made the necessary corrections to improve the manuscript. Below is a detailed response to the issues you have raised:
1.Thank you for your attentive review. At line 10, we have added a space to correctly format as "cadmium (Cd)" and have thoroughly checked the rest of the document for such oversights and made the necessary corrections.
2.Following your suggestion, at line 12, we have corrected the format of "Lolium perenne L." to ensure that "L." is not italicized. We have made the same correction for all species names in the manuscript, particularly at lines 58, 59, 61, 77, and 80. We have also reviewed and amended any such errors in the cited references, now highlighted for clarity.
3.Again, we appreciate your valuable input. At line 16, we have amended the format of "Peganum harmala L" by removing the italics from "L" and have standardized the format for all species names mentioned in the manuscript, adding or removing a period after "L" as required.
4.In response to your feedback, at line 16, we have changed "both arid sandy" to "both arid and sandy" to enhance clarity. Similarly, at lines 31-32, we have revised "absorbed and transported and accumulated" to "absorbed, transported, and accumulated" for better readability.
5.As per your recommended modification, at lines 33-34, we have altered "its stable, non-degradable characteristics" to "its stable, and non-degradable characteristics" to strengthen the grammatical structure of the sentence.
6.At line 36, we have added a period after "organelles," changing it to "organelles. Simultaneously."
7.At lines 52 and 54, we have corrected the format of "Solanum nigrum Linn" to ensure that "Linn" is not italicized. Thank you once again for your diligent review.
8.Following your advice, at line 59, we have removed the unnecessary word "soils."
9.At line 60, we have added a space to "B.subtilis" and formatted it as "Bacillus subtilis."
10.At line 72, we have changed "This study" to "this study" to maintain consistency in tone.
11.At line 83, we have provided the full description of the abbreviation "LB" upon its first use.
12.At your suggestion, at line 89, we have capitalized the first letter of "phylogenetic" to read "Phylogenetic."
13.We apologize for the oversight at line 95 where "16s rDNA" did not have the "s" capitalized. The correction has been made, and we have checked and amended all related formats throughout the text, thank you once again for your input.
14.At lines 98-104, we have applied the justified paragraph format as requested, and in accordance with the submission guidelines, we have adjusted and modified the relevant formats throughout the document to maintain the overall structure of the article.
15.At line 110, following your suggestion, we have changed "After 3 days" to "After three days."
16.At lines 112 and 114, we have described the abbreviations "MSA" and "CAS" upon their first use.
17.At line 140, we have added a space to "(1)FeCl3," correctly formatting it as "(1) FeCl3." We have also checked the entire document to ensure the correct format.
18.At line 154, we have corrected "bberellin" to "gibberellin" and apologize for the error, which has now been amended as per your suggestion.
19.At line 167, we have removed the unnecessary space between "37 °C," correctly displaying it as "37°C."
20.In response to your comments, at lines 239-240 and 314-319, we have applied the justified paragraph format to the sections.
21.At line 313, we have capitalized the first letter of "phylogenetic" to read "Phylogenetic."
22.At line 322, and lines 324 and 325, we have capitalized the "s" in "16S rDNA" and ensured that the gene names "gyrB" and "dnaK" are italicized.
23.At line 332, we have reformatted "36.2% and 49.9%" to "36.2 and 49.9%," removing the unnecessary percentage symbols.
24.At lines 400-402, we have revised the text for clarity and ensured the correct format for percentage values.
25.At line 418, we have corrected "P = 0.05 level" to "p = 0.05 level," changing "P" to lowercase "p."
Once again, we thank you for your attention to detail and appreciate the opportunity to enhance our manuscript. We believe these corrections have addressed the issues you raised and significantly improved the quality of our work.
Sincerely,
LingLing Wu

Reviewer 3 Report
Comments and Suggestions for Authors
Again, the abstract did not adequately express the research content.
How was the result of the bioactivity and heavy metal tolerance of the two strains? How was the identification of the two species-level identification of the two strains? What type of siderophores were produced by the two strains? What genes were related to the siderophore of strain CD3030?
Again the first paragraph of Discussion described general research background and had little direct relation to this research results. Preferably, discuss the fate of Cd in soil and the plant. As the result of chelating Cd by siderophores, Cd is uptaken to the plant? Explain in the Figure 10.
Linné, L. should not be described in Italic.
Comments on the Quality of English LanguageImproved.
Linné, L. should not be described in Italic.
Author Response
Dear Reviewer,
We extend our profound gratitude for the invaluable feedback and comments you have provided on our manuscript. Your constructive suggestions have been instrumental in enhancing the quality of our article, prompting us to approach the manuscript with greater diligence and precision. In response to your insights, we have thoroughly revised and refined our manuscript to improve the clarity and relevance of the abstract and discussion sections.
Regarding the bioactivity and heavy metal tolerance of the two strains, we have now elaborated in the abstract section of the article, detailing their impact on plant growth under heavy metal stress and their capacity to tolerate and potentially alleviate the presence of heavy metals in the soil. We have also clarified the species-level identification process of the two strains and described the siderophore types associated with them, with WL1210 being catechol-type and CD303 being carboxylic acid ferrophilin, ensuring that the methodology and results are presented with clarity. The focus of our study is on the growth effects of the two siderophore-producing Bacillus strains on ryegrass under heavy metal stress, and thus, we have provided a detailed description of the bioactivity assays in the Materials and Methods section, and discussed their implications for plant growth in the Discussion.
In line with the overall structure of the article, we have readjusted and modified the abstract section to meet your expectations. Should there be further modifications required for the abstract, we are prepared to make additional adjustments based on your valuable suggestions and feedback. We are immensely thankful for your guidance.
As for the genes related to siderophore production in strain CD303, we have expanded our discussion to include the identification of functional genes such as fnr and iscA, which are implicated in siderophore biosynthesis and promotion of plant growth.
We have also revised the first paragraph of the Discussion to focus more directly on the research findings, particularly the role of siderophores in the uptake of cadmium by plants and the implications for soil remediation and plant health. Following your advice, we have further discussed how heavy metals impact plant growth and physiology, and how beneficial microbes—such as Bacillus—can promote plant growth and mitigate heavy metal pollution. The modifications we have made are highlighted in the manuscript, and relevant literature is cited to support our assertions.
We have included an explanation for Figure 10 in the Discussion section and apologize for the lack of clarity in the figure's labeling, which may have caused confusion. We have revised and adjusted Figure 10 to illustrate the following: the extracellular polymeric substances (EPS) of Bacillus species are rich in reactive functional groups that bind to heavy metal ions, immobilizing them on or within microbial cells, thereby reducing the environmental impact of these contaminants. Additionally, siderophore-producing Bacillus strains secrete a variety of extracellular metabolites, including organic acids, proteins, and siderophores, which can chelate and precipitate heavy metal ions, diminishing their bioavailability and mobility. Furthermore, Bacillus species can alter the chemical speciation of heavy metals through metabolic processes, such as the reduction of hexavalent chromium (Cr (VI)) to trivalent chromium (Cr (III)), thereby reducing the metal's toxicity. Concurrently, the symbiotic relationship between siderophore-producing Bacillus and plant roots promotes plant growth and enhances the plants' ability to absorb and accumulate heavy metals.
Lastly, we have corrected the formatting error for Linné, L., ensuring that it is no longer presented in italics.
We believe these changes have significantly improved the manuscript, and we are grateful for the opportunity to refine our work based on your insightful comments.
Sincerely,
LingLing Wu
Reviewer 4 Report
Comments and Suggestions for Authors
The authors have introduced all required modifications and corrections into the manuscript. The current version of the manuscript has been essentially improved, and now it is suitable for publication.
Author Response
Dear Reviewer,
We are immensely grateful for your constructive feedback and the time you have dedicated to reviewing our manuscript. Upon receipt of your positive feedback, we were truly elated and appreciative of the recognition and support you have provided for our research. In response to the valuable input from experts such as yourself, we have meticulously reworked and enhanced our manuscript to meet the publication standards. We are deeply grateful for your constructive comments and express our sincere thanks once again for your encouragement.
Sincerely,
LingLing Wu
